# Self-vectoring electromagnetic soft robots with high operational dimensionality

Wenbo Li [1,2,4] ✉, Huyue Chen [3,4], Zhiran Yi[1], Fuyi Fang[1], Xinyu Guo[1], Zhiyuan Wu [1], Qiuhua Gao[1], Lei Shao [3] ✉, Jian Xu[2], Guang Meng[1] & Wenming Zhang [1] ✉

Soft robots capable of flexible deformations and agile locomotion similar to biological systems are highly desirable for promising applications, including safe human-robot interactions and biomedical engineering. Their achievable degree of freedom and motional deftness are limited by the actuation modes and controllable dimensions of constituent soft actuators. Here, we report self-vectoring electromagnetic soft robots (SESRs) to offer new operational dimensionality via actively and instantly adjusting and synthesizing the interior electromagnetic vectors (EVs) in every flux actuator sub-domain of the robots. As a result, we can achieve high-dimensional operation with fewer actuators and control signals than other actuation methods. We also demonstrate complex and rapid 3D shape morphing, bioinspired multimodal locomotion, as well as fast switches among different locomotion modes all in passive magnetic fields. The intrinsic fast (re)programmability of SESRs, along with the active and selective actuation through self-vectoring control, significantly increases the operational dimensionality and possibilities for soft robots.

Soft robots inspired by biological systems with excellent agility, adaptability and safety have shown great potential in search-and-rescue, safe human-machine interaction, and biomedical applications[1–3]. Their capabilities of deformation and locomotion are dominantly determined by soft actuators which remain the main bottleneck for soft robotics. Specifically, soft robots have infinite degrees of freedom in theory but only limited numbers of actuators in practice, and thus more sophisticated motion and shape morphing will require more actuators and more cumbersome control systems[4,5]. Despite diverse soft actuators controlled by light, heat, humidity, pressure, and electric or magnetic fields have been developed, their integration level and actuation capacity in the underactuated soft robotic systems are still far from biological systems[6–10].

Generally, one actuator can only control one mode or direction of deformation and motion for soft robots[1,5,11], including dielectric elastomer actuators, HASEL, shape memory alloys or polymers, etc. They typically can only realize one actuation mode like elongation, contraction, bending or twisting by each specific structure or design[12]. Raising the number of actuators or introducing variable stiffness mechanisms can help to increase the actuation modes[13,14]. Likewise, introducing additional control sources is another solution to enrich the actuation modes for a limited number of actuators. For example, fluid elastomeric actuators can be actuated respectively by inflation pressure and vacuum to achieve two opposite deformations[15]. Hydrogels or polymers embedded with multifunctional particles can be activated simultaneously by light and magnetic field, thus achieving multimodal actuation in one actuator entity[10,16]. However, these approaches lead to more complex soft structures and control apparatus. As a result, achieving more agile actuation modes and diverse deformations while maintaining the ease of deployment is still a well-known challenge.

[1]State Key Laboratory of Mechanical System and Vibration, School of Mechanical Engineering, Shanghai Jiao Tong University, Shanghai 200240, China. [2]School of Aerospace Engineering and Applied Mechanics, Tongji University, Shanghai 200092, China. [3]University of Michigan–Shanghai Jiao Tong University Joint Institute, Shanghai Jiao Tong University, Shanghai 200240, China. [4]These authors contributed equally: Wenbo Li, Huyue Chen. ✉e-mail: wenboli@tongji.edu.cn; lei.shao@sjtu.edu.cn; wenmingz@sjtu.edu.cn

Remarkably, the recently thriving soft magnetic robots, usually built by soft materials embedding pre-polarized magnetic particles, demonstrate a feasible way to overcome the above challenge by achieving excellent maneuverability and flexibility close to biological systems[8,9]. Due to the advantages of wireless control, fast response, easy miniaturization, and magnetic (re)programmability, soft magnetic actuators and robots open a broad prospect for future biomedical applications[17–19]. Nevertheless, the present control systems of external magnetic field for realizing multimodal locomotion and 3D shape morphing[8,9,20,21] are complex and bulky, and also very difficult to only target particular domains of a robot for actuation. More importantly, the reported reprogramming methods to modify the polarization of magnetic domains in soft materials based on heating, are slow, tedious and impossible for in-situ customization[22–24]. To solve these issues, electromagnetic actuators have lately been noticed as a promising alternative by replacing magnetic domains in soft materials with conductors passing electric currents, which can be easily controlled to generate a desired Lorentz force in response to a constant external magnetic field[25–27]. Such actuators show advantages of fast reprogrammable control, domain selective actuation, and MRI-compatibility upon soft magnetic systems. Besides, recent designs based on liquid-metal (LM) coils embedded in soft shells further demonstrate excellent stretchability, compliance and biocompatibility[28–31]. Yet, the envisioned opportunity of rich electromagnetic vector (EV) composition and control has not been fulfilled, and it provides a potential pathway to achieve unprecedented actuation and controllable dimensions for soft robots.

Herein, we introduce a new concept of self-vectoring control for magnetic actuation. The proposed self-vectoring electromagnetic soft robots (SESRs) demonstrate previously unachievable shape morphing and locomotion capabilities in soft magnetic and electromagnetic robots. Via the reconfiguration of two kinds of elementary actuator modules (a soft vertical flux actuator and a horizontal flux actuator), SESRs with various shapes, EV directions are fabricated. Such a design principle allows the self-synthesis of EVs through synchronous input currents carried by different modules of a SESR, leading to any arbitrary magnetic vectors or their arrays. As a result, the fast tuning and reversing of input currents in each module enable selective, independent reprogrammable actuation and instant omnidirectional vector synthesis of all sub-domains in SESRs, which has not been achieved in previous soft robots. With this novel design and self-vectoring control, we achieved unmatched actuation capabilities in soft robots with limited number of actuators and signals, such as omnidirectional rotation and rolling, in-plane free orientation, complex multimodal locomotion, sophisticated and instant 3D shape morphing. More specifically, we show agile rolling locomotion on rugged terrains, multi-limb synergistic dancing rhythmically of a trefoil-shaped robot, and other bioinspired locomotion, crawling, flipping, and untethered paddling with differently configured SESRs. Furthermore, switching between some of these locomotion modes, along with different moving directions, is also realized by reprogrammable shape changes in just one SESR robot, revealing its exceptionally high control dimensionality.

## Results
### Principles of design and self-vectoring control
Magnetic actuation is usually implemented through manipulating permanent magnets, electromagnetic coils, or electromagnets to generate spatially vectoring fields. The magnetic torques and forces exerted on magnetic robots can control their deformation and motion in the workspace. Here, we propose a distinction between external and interior magnetic actuation for vector control (Fig. 1a). We define the aforementioned magnetic actuation of changing the applied field

direction as external vector control. The new concept of self-vectoring control is to actively and instantly adjust and synthesize the interior vectors in every sub-domain of the robots, thus to realize active and selective actuation in passive and constant fields. As a result, dual vector control can be explored to manipulate magnetic robots in higher dimensionality. We conceive two elementary modules with interior reprogrammable EVs, module V contains a vertical electromagnetic flux $\mathbf{B_V}$, and a horizontal flux $\mathbf{B_H}$ corresponding to module H. As we know, two orthogonal EVs can generate any vector in a plane following the parallelogram law of vector synthesis. Further, arbitrary spatial vector $\mathbf{B}(r, \theta, \varphi)$ can be synthesized by three orthogonal EVs in theory as shown in Fig. 1a. Then through different combinations of the two elementary modules, vectors in three planes can be synthesized (Fig. 1b), B$(r, O, \varphi)$ and $\mathbf{B}(r, 90°, \varphi)$ by modules V and H, $\mathbf{B}(r, \theta, 90°)$ by two modules H. When subject to a passive and constant external magnetic field, the deformation and motion of the modules can be actuated arbitrarily by the magnetic torque and force exerted on the synthesized vectors.

To build a complete soft solution for self-vectoring actuation and robots, we design two kinds of soft flux actuator modules corresponding to two typical electromagnetic coils (planar spiral and 3D helical). As shown in Fig. 1c, two-layered liquid metal coils embedded in quadrate silicone elastomer matrixes constitute the soft flux actuators. The rectangular spiral LM coil corresponds to the soft vertical flux actuator (module V), while a unique reconfigurable helix LM coil is designed for the soft horizontal flux actuator (module H). Multiple parallel LM coils connect in sequence and encompass an inflatable chamber to form the reconfigurable helix LM coil. The chamber is located in the middle layer of the elastomer matrix and inside the helix coil, which can be inflated by air or fluid to reconfigure the helix coil and the actuator shape. We can pre-store a volume of air in the chamber or inject different volumes through a connected tube whenever necessary.

The orientation and magnitude of the EVs generated by LM coils can be controlled by input currents, following Ampere's right-hand screw rule and Biot-Savart's law. Moreover, both ends of the double layered liquid metal coils are designed to converge at the same corner of the actuators and connect to the power source through soft twisted-pair wires. By this method, the effect of the tethered control on the actuators when subject to external magnetic field can be minimized. The detailed structure and fabrication process of the two types of soft flux actuators are depicted in Supplementary Figs. 1–6. The module H also has a very good compliance and robustness after inflation, which can be stretched, twisted, or pressed to a large extent (see Supplementary Fig. 7). The fabricated module samples have a good consistency, for their resistance and inflated outer profiles with different inflating volumes vary in small scale as shown in Supplementary Fig. 8. The temperature variations of the H samples with different inflating volumes under different current signals are also characterized, the results show that the temperature of the samples eventually tends to be stable (see Supplementary Fig. 9). The heating is slightly more pronounced for larger inflating volume, mainly caused by the increase of coil resistance with the inflating volume for module H.

The quadrate shape of the two actuator modules allows them to assemble and reconfigure expediently in parallel or in series to form new modules and configurations. As shown in Fig. 2a, a module H actuator stacks up with a module V or H to constitute two composite modules V‖H and H‖H, respectively. The two modules are bonded together after assembly, then inflating the module H to result in the bulging configurations. The corresponding simulations of synthetic electromagnetic fields are illustrated in Supplementary Figs. 10 and 11. A soft Halbach Array with three H modules and two V modules aligning in series alternately was also fabricated, which can be inflated or folded to a more compact arrangement as shown in Fig. 2a.

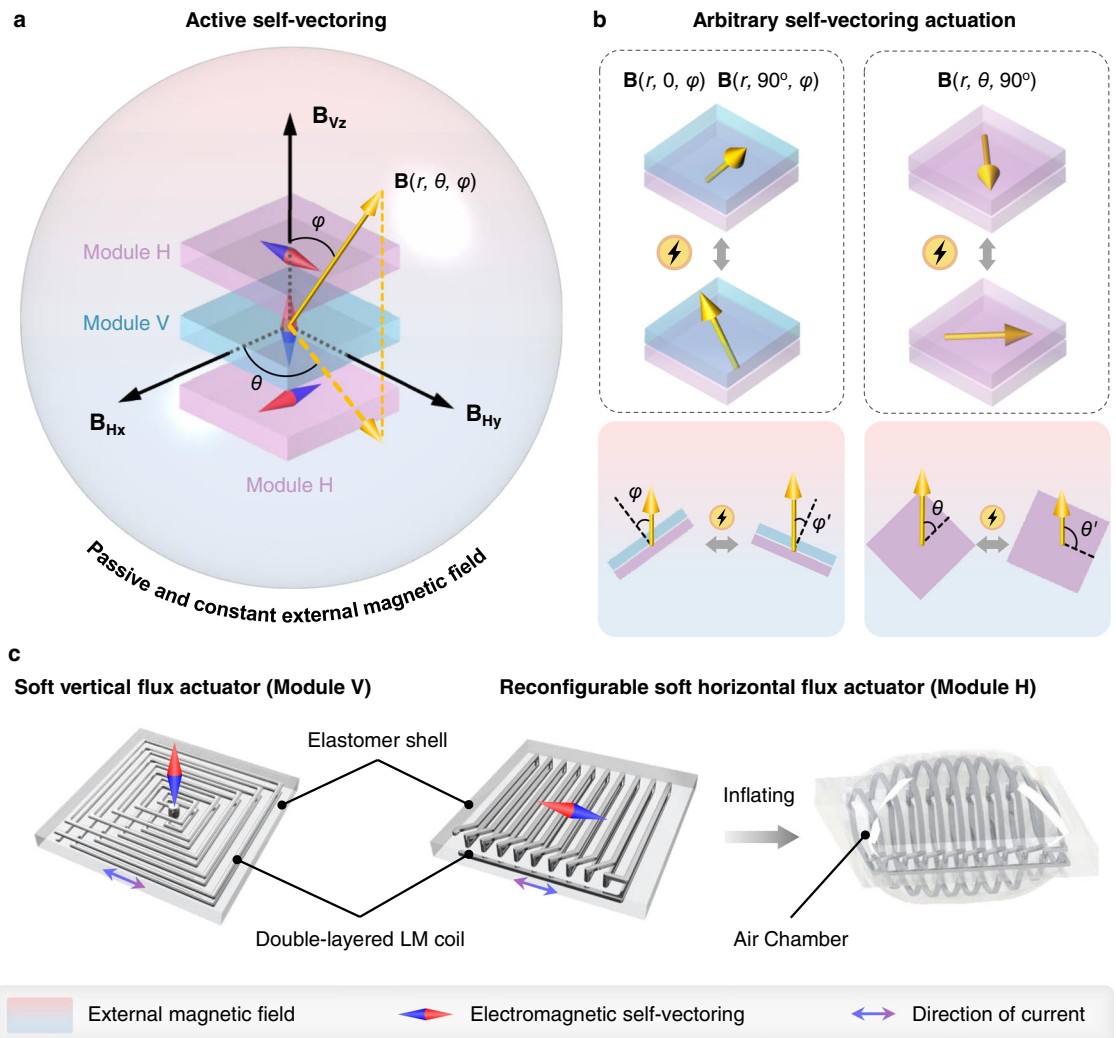

**Fig. 1 | Concept of self-vectoring and the elementary soft actuator modules of SESRs. a** Schematic of active self-vectoring of the SESRs. The sphere space with gradient from blue to red represents the passive and constant external magnetic field. Three modules (two elementary modules: module V with a vertical EV, module H with a horizontal EV) in the sphere space can synthesize any EVs in theory. **b** Design principle of self-vectoring control for SESRs. Combination of two elementary modules can constitute new composite actuators with arbitrary EVs as applying current signals can easily and instantly adjust the interior EVs in every sub-domain of the robots, thus resulting arbitrary self-vectoring actuation in passive and constant external magnetic fields. **c** Soft solution of the two elementary soft electromagnetic actuator modules consisting of double-layered liquid metal coils embedded in elastomer shells. The soft vertical flux actuator (Module V) has a square spiral coil. The reconfigurable soft horizontal flux actuator (Module H) has a built-in chamber which can be precharged or inflated by applied pressure to reconfigure the helix coil and the actuator shape. The directions of EV can be determined by the right-hand rule, while EV synthesis follows the parallelogram law.

## Validation and demonstration of the self-vectoring control

To validate the self-vectoring characteristics of the actuators, we placed the composite modules V∥H and H∥H in an external magnetic field and applied different currents on each elementary module (Fig. 2b, c). Because current-carrying wires are subjected to Lorentz force under a magnetic field, a net Lorentz torque can be induced for a closed loop of wires. Thus, the actuators deform and move under the action of the driving force $\mathbf{F} = I \cdot \int d\mathbf{l} \times \mathbf{B}$ or the torque vector $\mathbf{T} = nI \cdot (\mathbf{A} \times \mathbf{B})$, where $I$ is the applied current, $d\mathbf{l}$ is the infinitesimal length vector of the coil, $n$ is the number of coil loops and $A$ is the area plane vector of the loops[25].

Figure 2b illustrates the high-dimensional and reprogrammable shape morphing of a trefoil-shaped SESR which consists of three V∥H modules connected by a middle triangular elastomer matrix. The three coils in modules V and H are respectively connected in series and controlled by two current signals $I_1$ and $I_2$ (Supplementary Fig. 12). We demonstrate multiple morphing modes and states of the trefoil-shaped SESR on a plate magnet using different control sequences with

varying amplitude, frequency, and polarity (Supplementary Movie 1). Three 'leaves' twist on the surface from side to side when only applying bipolar square-wave signal with 1.8-A amplitude to the modules H in series, 1.2-A DC signal applied on the modules V controls the 'leaves' to bend. Moreover, when applying the two current signals simultaneously, three 'leaves' twist and bend rhythmically like dancing.

Figure 2c shows the agile rolling locomotion of a single H module and its composite module H∥H. First a single module H acts as a rolling soft robot which can be controlled by only one current signal to roll continuously on different rugged terrains (also see Supplementary Fig. 13, Movie 2). Then a composite module H∥H demonstrates the omnidirectional rolling controlled by only two current signals. Here we give eight representative synthesized rolling directions corresponding to eight synthesized EVs with various directions. In addition, programming eight groups of pulse current signals in a sequence can control the module to rotate around the center for a rotary lighting operation (Supplementary Figs. 14, 15 and Movie 3). This demonstrates that synthesis and reprogramming of EVs controlled by only two

## a  Electromagnetic vector synthesis of soft flux actuators

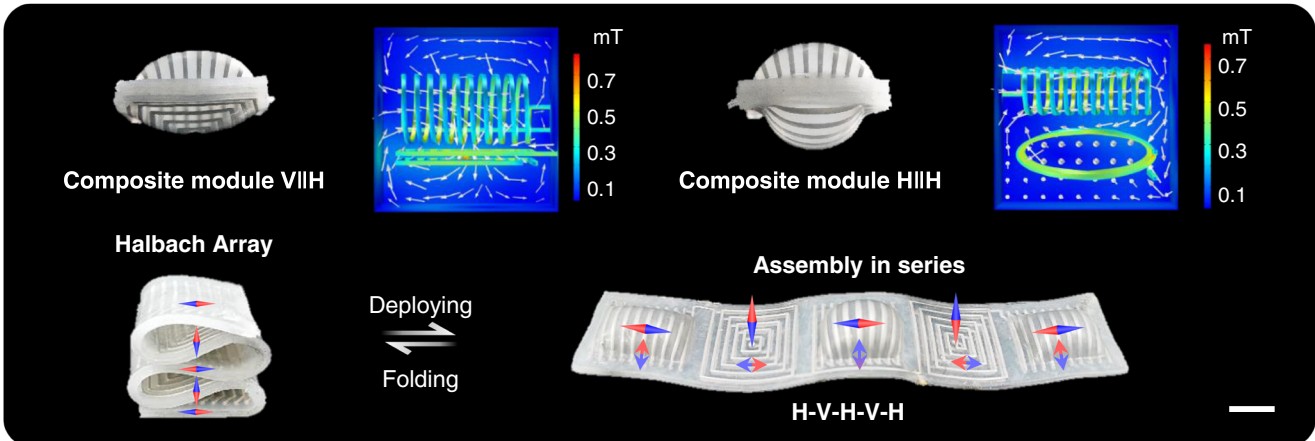

### Self-vectoring control under a constant magnetic field

## b  High-dimensional shape morphing

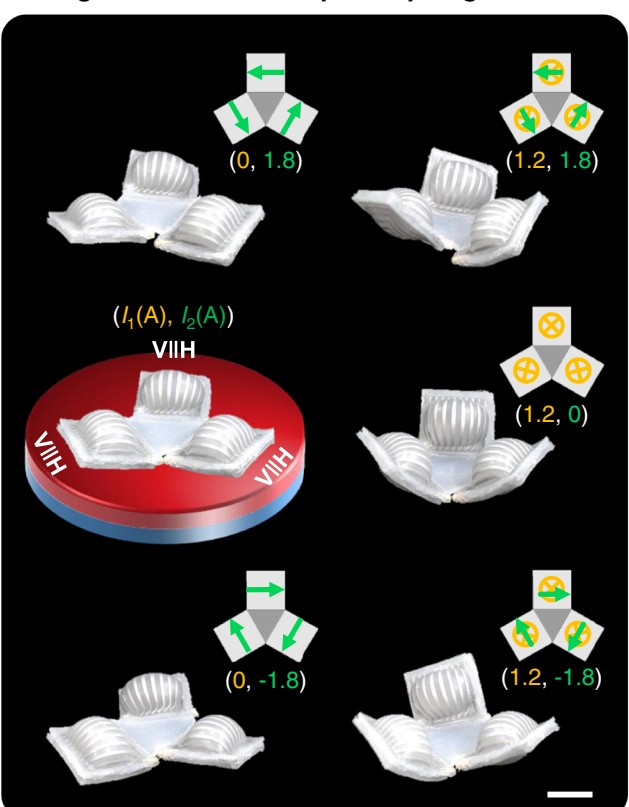

## c  Agile rolling locomotion

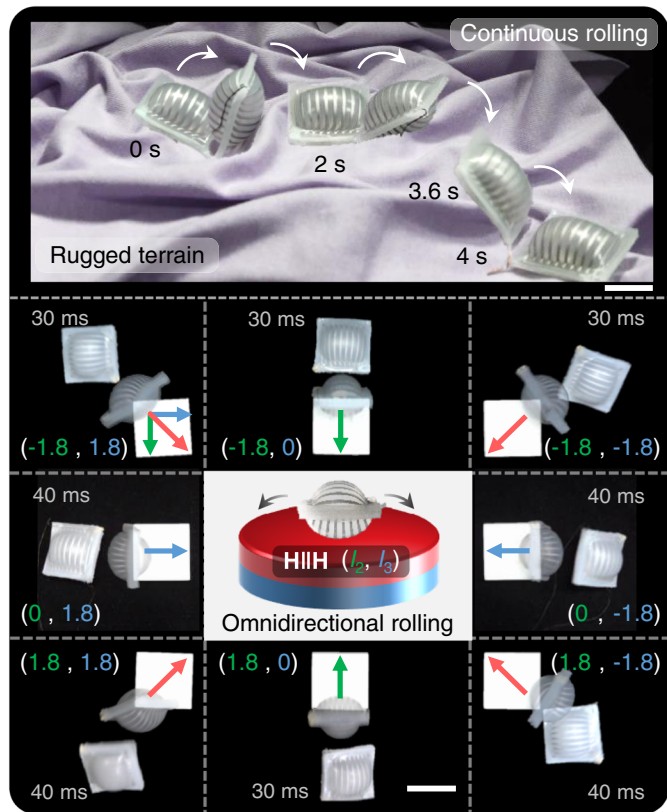

**Fig. 2 | Different configurations of SESR prototypes and self-vectoring control under a constant magnetic field. a** EV synthesis and different combinations of the two soft flux actuator modules. Combining module H with module V or another module H in parallel can generate two elementary composite modules, referred as V||H and H||H, respectively. A soft Halbach Array is fabricated by connecting multiple H and V modules alternately in series, which can be folded and deployed. Scale bar = 10 mm. **b** High-dimensional shape morphing of a trefoil-shaped SESR

assembled by three composite modules V||H in parallel. Every 'leaf' can demonstrate three modes of deformations including bend, twist, bend and twist under control of two signals. Scale bar = 20 mm. **c** Demonstration of the agile rolling locomotion of a single H module and its composite module. A single module H can roll continuously on rugged terrains controlled by only one current signal, scale bar = 20 mm. The composite module H||H can realize omnidirectional rolling on the magnet surface controlled only two current signals, scale bar = 30 mm.

current signals can lead to more than two rolling directions and shape morphing modes for the SESRs.

### Actuation characteristics
To understand the principles of shape morphing and movement of the SESRs, we analyze the force and motion of the elementary modules V and H in an ideal parallel magnetic field for general mechanism study (as shown in Fig. 3a, b). The direction of the module's EV tends to align with the same direction as the external magnetic field, due to the torque $T$ generated by Lorentz force if ignoring gravity and elastic deformation of the module. Figure 3c, d show the in-plane free orientation operations of suspended modules between two blocky magnets for verifying the self-vectoring control. The characterization of magnetic field, experimental setup and method are provided in

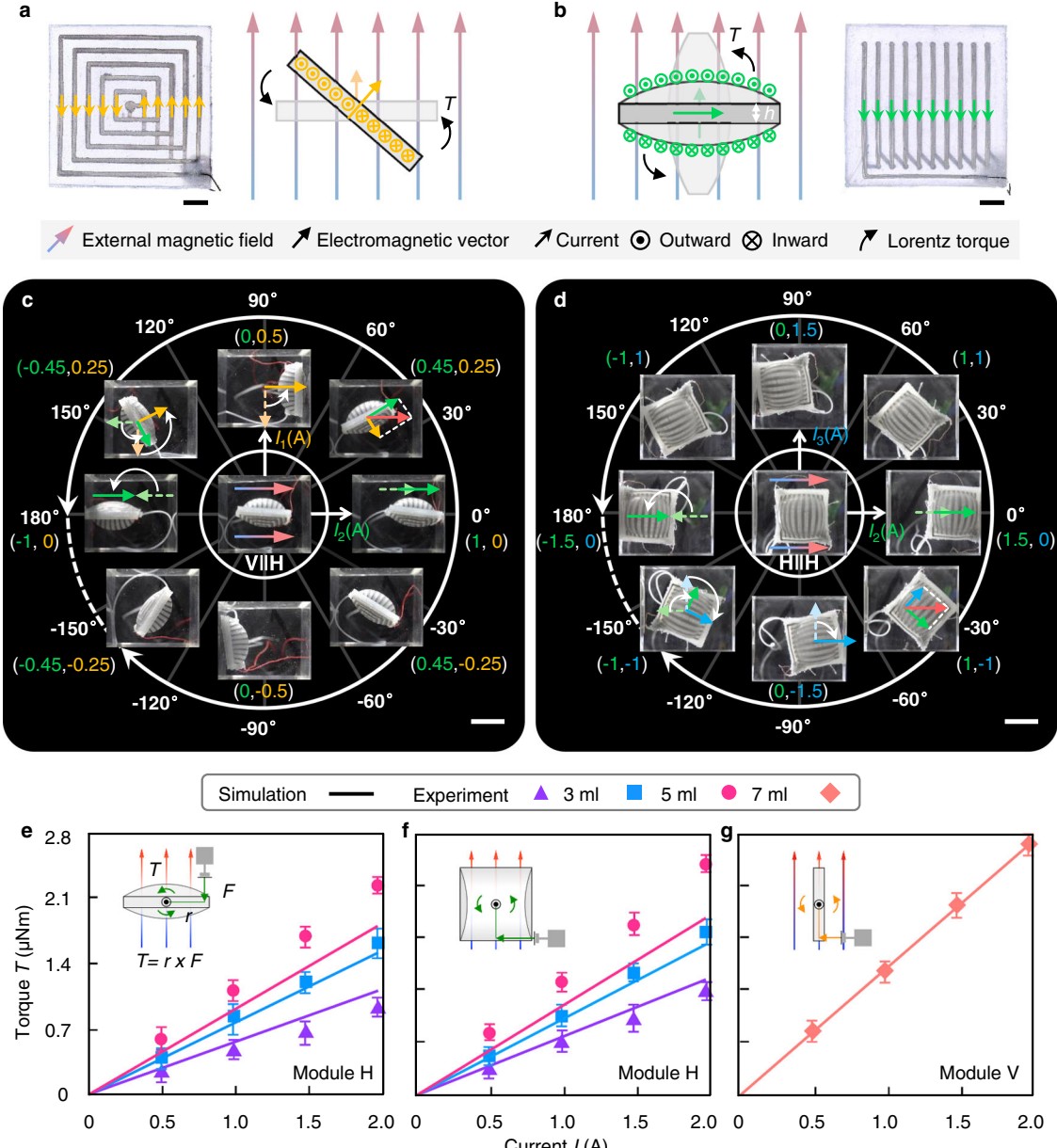

Fig. 3 | Validation and characterization of self-vectoring control for SESRs.
**a**, **b** Analysis of Lorentz force and motion for the two elementary modules V and H in an ideal parallel magnetic field, respectively. Scale bar = 5 mm.
**c**, **d** Demonstration of orientation and self-vectoring control for the modules V||H and H||H in an approximately parallel magnetic field (between two parallel magnets with opposite polarities), respectively. Scale bar = 20 mm. **e**, **f** Comparison of experiment and simulation on the maximum Lorentz torque of the module H. The module is hung in the constant magnetic field with different posture for (**e**) and (**f**). **g** Comparison of experiment and simulation on the maximum Lorentz torque of the module V varying with applied current. Error bars represent standard deviation.

Supplementary Figs. 16 and 17. We can see either only one elementary module V or H is activated, or two modules are controlled synchronously to synthesize a new EV according to the right-hand rule and the parallelogram law of vector synthesis, and composite modules move straightly to the right or rotate by certain angles to make the directions of the EVs consistent with the external magnetic field. We respectively demonstrate eight rolling motions towards eight directions by different control signals for the two composite modules. Due to the fact that module H||H is heavier and the rotational resistance from tubes and hanging cable is more significant than that of module V||H, a larger current was needed to rotate the module H||H.

We then investigate the Lorentz torque of the two elementary modules experimentally and numerically in the same magnetic field, as shown in Fig. 3e–g. When the direction of the coil section (following the right-hand rule) is perpendicular to the constant magnetic field, we

obtain the maximum torque corresponding to different currents as shown in the figures. The torque increases approximately linearly with the current, and for module H, larger inflating volume results a higher torque. The simulations show good agreement with the experimental results, with the deviations mainly coming from the simplified simulation model of the LM coil and the measuring error. We also evaluated the Lorentz torque when the modules were hung in the magnetic field at 45° angle (see Supplementary Fig. 18). The detailed methods of simulation and characterization can be seen in Supplementary Notes and Supplementary Fig. 19.

When placing the modules on a plate magnet as demonstrated in Fig. 2, the effects of gravity and variation of magnetic field on the modules' actuation characteristics must be considered. The deformation and movement of module V with different constraint conditions in an axisymmetric magnetic field have been

analyzed in detail by Mao et al.[28] Here we focus on the newly proposed module H which has a variable longitudinal profile of approximately elliptic. We characterized the magnetic field distribution (Supplementary Fig. 20) and investigated different current signals and inflating volumes on the actuation characteristics of several module H samples. The detailed experimental setup and methods can be found in Methods and Supplementary Fig. 21.

Figure 4a–d demonstrate four different modes of motions for the module (Supplementary Movies 4 and 5). They are corresponding to different inflating rates of air and current signals. Specifically, Fig. 4a is the inflating and rotation process when the module is inflated by different air flow under a certain current. Figure 4b is the side-to-side swaying motion when the module is subject to different AC current signals with slow loading rate. Figure 4c shows the side-to-side flipping motion when the module is excited by periodic pulse or step current signals. The EV flips together with the module, thus the periodic current with unipolarity can induce alternating clockwise and counterclockwise Lorentz torques which drive the module to flip side-to-side on the magnet. However, if changing the current polarity in the signal period, the direction of the EV reverses from right to left as the initial state, then a same clockwise torque makes the module flip right again to realize a continuous flipping motion as shown in Fig. 4d.

Figure 4e shows the force and motion analysis of the module H. The rotation motion is determined by a combination of Lorentz forces ($F_{Li}$ and $F_{Ri}$), Gravity ($Mg$), reaction force ($F_N$), and frictional force ($f$). The detailed analysis and derived process are given in Supplementary Notes. We obtain the theoretical relation between the rotation angles and applied current which is also verified by experiments as shown in Fig. 4f. The rotation angle of the inflated module first increases with the current, and larger inflating volume corresponding to larger angle under the same current. Then, due to the fact that practical profile of the inflated module is different from the hypothetical ellipse, the convex parts on both sides lead to the change of tendency on the rotation. When the endpoint of the convex part begins to contact the surface, the rotation angle will stay at this critical angle $\theta_c$ with the increasing of current, until big enough current rotates the module again. However, during the practical rotation process, the soft convex part deforms and bends under gravity, leading a slight discrepancy from the theoretical analysis. More experimental results about the steady rotation of the module H are provided in Supplementary Figure 25.

We also investigated the dynamic rotation process of the module H. The dynamic rotation speed $v$ is evaluated by the average speed from 0 to 90 degrees, showing that the sample with the same air volume rotates faster under higher current (Supplementary Fig. 26a). When applying a high current (2 A), the speed also increases with the air volume, as shown in Fig. 4g. The maximum speed corresponding to $V_a = 7$ ml reaches about 26.7 rad s$^{-1}$, and the dynamic response process is plotted in the inset. In response to a 2-A step current signal, the module rotates about 176° in 0.14 s, and then gradually approaches to a steady angle of about 132° after 1.2 s. In addition, the effects of inflating rate on the rotation motion of module H were also investigated. Supplementary Fig. 26b, c show lower inflating rate leads to larger transient angle under the same volume, because it takes longer to achieve the same air volume by lower inflating rate, and the rotation angle is thus closer to the steady state. It can be concluded that higher current results in a larger Lorentz torque, and thus a larger angle response.

We further investigated the actuation characteristic of module H under different current waveforms. The rotation angle in response to triangular-wave signals with different frequencies ($f = 0.5, 1, 2$ Hz) are plotted in Supplementary Fig. 26d, showing that the angles basically vary with the signals. Higher frequency leads to larger angular response amplitude, which is mainly because the dynamic effect is more pronounced at high frequency for the rotation. The periodic

swaying motions in response to triangular-wave signals with different amplitudes for modules with different air volumes are also shown in Supplementary Fig. 26e, f. We also tested the motion responses to different waveforms for the module, as shown in Fig. 4h, a triangular-wave signal and a square-wave signal with the same amplitude $I = 1$ A and frequency $f = 0.5$ Hz lead to distinct motion responses. The triangular-wave signal has a slow loading rate, thus gently actuating the module to raise and lie down as a quasi-static motion. In contrast, sudden rising edge of the square-wave signal drive the module to flip over 90° clockwise and finally lie on the magnet surface after removing the current in the second half period. When subject to a bipolar square-wave signal with amplitude $I = 1.5$ A and frequency $f = 1.25$ Hz, the module rolls continuously as shown in Fig. 4d. The continuous flipping angle and the displacement of center of mass are depicted in Fig. 4i, j, respectively. We note that only three complete flips were observed due to the limited magnetic field range of the plate magnet, and the average speed reaches about 103.5 mm s$^{-1}$.

## Reprogrammable shape morphing

After characterizing the elementary modules of SESRs and validating their self-vectoring control, we then demonstrate the flexibility and scalability of these module actuators by different combinations of them and ample reprogrammable shape morphing. Figure 5a shows two H modules assembled in series and fixed in the middle to form an H-H structure. The two coils in the modules can be connected to two current signals for independent control or in series for synchronization control by one signal. The directions of the two synchronized EVs can be consistent or opposite, determining by two different configuration types of the coils. The positive current induces a clockwise moment which makes the left module raise and the right module collapse to the support. Reversing the current leads to the opposite deformations for the two modules. When applying bipolar square-wave signals with 1.5-A amplitude and different frequencies, we observed periodic swaying and flapping deformations of the structure corresponding to different series modes (Supplementary Movie 6).

Similarly, we assembled three H modules in series and demonstrate four groups of shape morphing for the reconstructed H-H-H strip (Fig. 5b, Supplementary Movie 7). They are programmed by three independent control signals which can be easily and rapidly changed or reversed. Figure 5c shows a parallel assembly for three H modules by a triangular silicone matrix and the achievable configurations (Supplementary Fig. 27 and Movie 7). Increasing the number of H modules can obtain more exciting combinations and plentiful shape morphing. Figure 5d shows three different combination types of a square structure assembled by four H modules at the center. The coil axes of the four modules are respectively parallel or perpendicular to each other, or perpendicular in diagonal modules. Four control signals with different amplitudes, frequencies and phases can be programmed to achieve rhythmed and interesting shape morphing (Supplementary Fig. 28 and Movie 8).

Figure 5e shows a composite assembly of two H modules and two V modules, or two V||H modules connected by a square linking module to form a strip structure. Unlike the serial assembly, the EVs of H modules are perpendicular to the strip. Therefore, we can realize twisting, bending and folding deformations of the strip by different self-vectoring control (Supplementary Fig. 29 and Movie 9). A more complicated composite assembly of V||H modules and the corresponding reprogrammable shape morphing have been demonstrated in Fig. 2b. By controlling the synthesis of EVs, more complex and ample 3D shape morphing can be obtained compared to structures constructed by single type of modules. Figure 5f demonstrates the programmable transformation and flipping locomotion of a serial structure V-H-V. Controlling by the programmed current signals, the V modules on both sides first fold to wrap the middle H module in

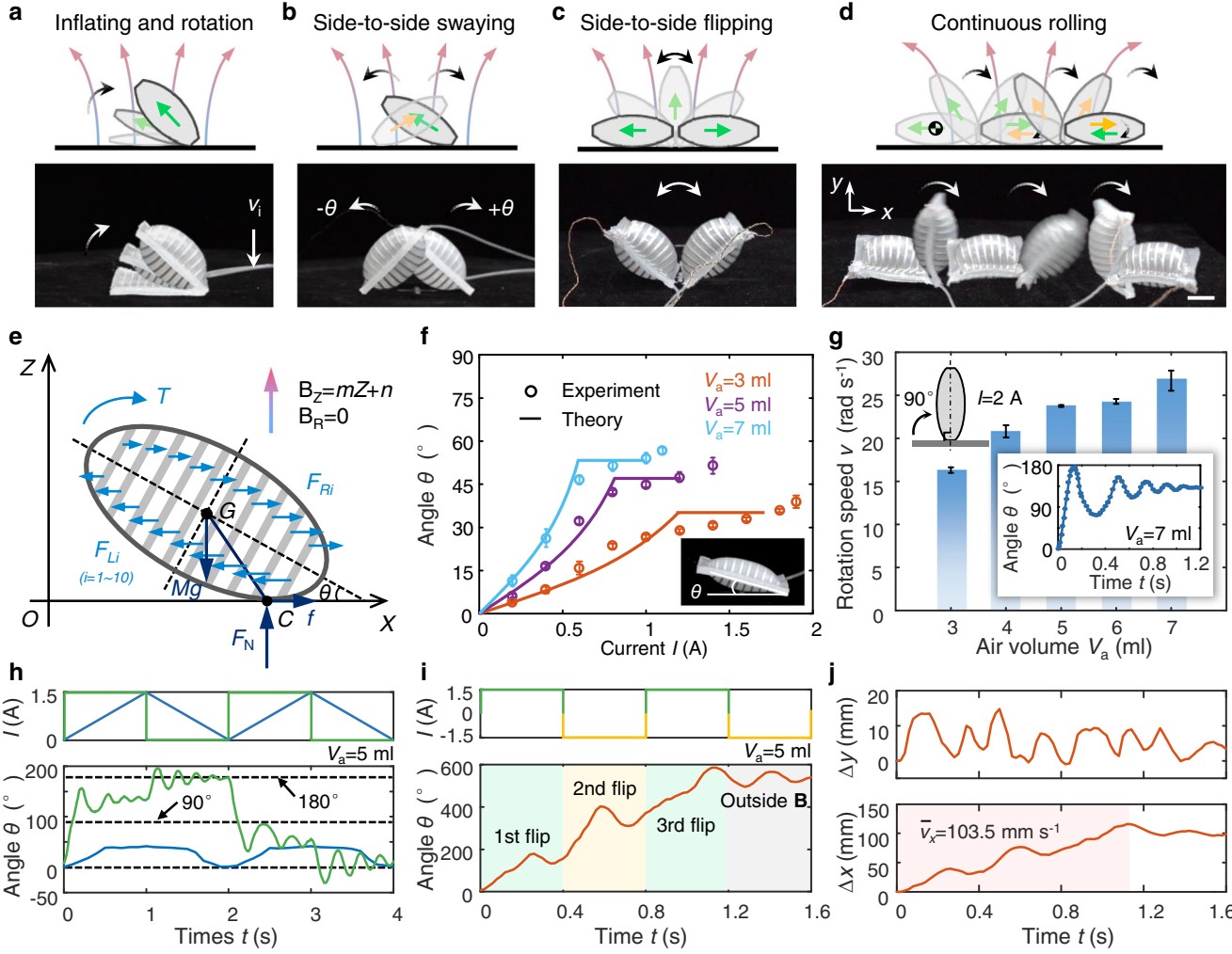

**Fig. 4 | Actuation characterization of the module H under a constant magnetic field. a** Inflating and rotation motion of the module when injected with air under a constant current. **b** Side-to-side swaying motion of the module when subject to AC triangular-wave current signals. The clockwise rotation angle is defined as positive, while the counterclockwise is negative. **c** Side-to-side flipping motion of the module when subject to unipolar square-wave signals. **d** Continuous rolling motion of the module when subject to bipolar square-wave signals. The center of mass is marked. Scale bar = 10 mm. **e** Force and motion analysis of the module H. **f** Comparison of theoretical and experimental results on the stable and critical rotation angle $\theta$ varying with the input current for different precharged air volume $V_a$. Error bars represent standard deviation. **g** The average rotation speed $v$ varying with the precharged air volume when the module flipping over 90° (a 2-A step signal applied). The inset shows he dynamic response of rotation angle for the module sample with $V_a = 7$ ml when excited by a 2-A step signal. Error bars represent standard deviation. **h** The comparison of angle response for the module when subject to periodic current signals with different waveforms. **i** The angle response of continuous rolling motion for the module when subject to a bipolar square-wave signal. **j** The displacement curves of the center of mass for the continuous flipping motion of the module.

sequence, then the reconstructed wheel-like structure flips to the left (Supplementary Movie 10).

## Multimodal locomotion

From the above characterization and demonstration, we know that single module H or composite modules can be used to achieve agile rolling locomotion. Through adjusting the control signals, we can also realize multimodal locomotion for different configured soft robots on a static magnet surface. Figure 6a shows two H modules are connected in series to constitute a crawler. Applying a bipolar square-wave signal to one module can make it oscillate side-to-side similar to the flapping motion shown in Fig. 5a. The other module acts as a passive and compliant tail which can convert the oscillation of the robot to a caterpillar-like crawling due to the differential friction. Therefore, selective actuating a different module H can make the robot crawl towards two directions (Supplementary Movie 11). The crawling mechanism and the speed variation with respect to the driven frequency are depicted in Fig. 6b. There is a

maximum crawling speed ~28.8 mm s⁻¹ corresponding to the 6-Hz and 1.5-A control signal.

When two H modules are connected in parallel, an H||H robot which can make flexible turning locomotion is obtained. Controlling the two parallel EVs to be opposite with 4-Hz and 1.5-A positive or negative square-wave signals, the robot can turn clockwise or anticlockwise with an average speed of 4.7° s⁻¹. (Supplementary Fig. 30a, Movie 12). Further, we demonstrate the crawling and turning locomotion with an instant transition by a soft robot composited by three H modules (Supplementary Fig. 30b, Supplementary Movie 13). Actuating the upper H module or the bottom two modules H can realize crawling or bi-directional turning in one robot, respectively. Moreover, a SESR can be constructed with an H-module head and a V-module tail, resulting in a robot can crawl, flip, as well as switch the locomotive modes by folding shape morphing (Fig. 6c). Such kind of a right-headed crawling motion is achieved by oscillation of the inflated H head, together with the passive V tail same as the H-H robot shown in Fig. 6a. Interestingly, the V tail can also fold onto its head,

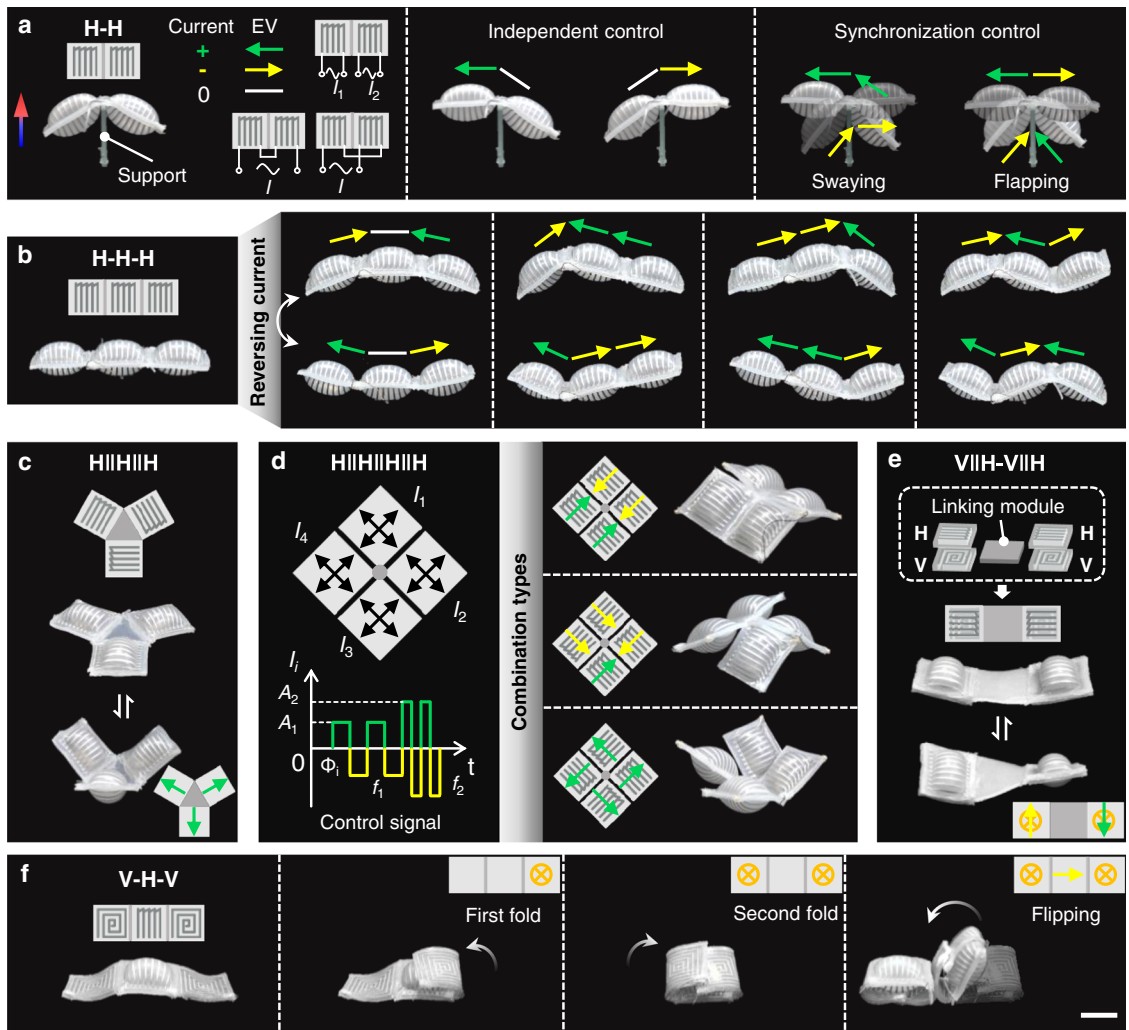

**Fig. 5 | Reprogrammable shape morphing of SESRs by self-vectoring control.**
**a** Independent and synchronization control of a symmetric H-H structure. The two modules can be controlled separately by two independent signals or by one signal as their LM coils can be connected together in series (either head-to-head or head-to-tail). **b** H-H-H, a strip structure assembled by three H modules in series. **c** H||H||H, a triangle structure assembled by three H modules in parallel using a central linking module. **d** H||H||H||H, three typical square structures assembled by four H modules with different orientations. **e** V||H-V||H, a strip structure assembled by two composite modules V||H in series with a middle linking module. **f** V-H-V, a strip structure assembled by one middle H module and two side V modules in series. Scale bar = 20 mm.

and then flip over and crawl towards the opposite direction (Supplementary Movie 14).

Based on the above crawling locomotion mechanism, we can also control a trefoil-shaped robot to realize the omnidirectional crawling (Fig. 6d, Supplementary Fig. 31 and Movie 15) and a tortoise-like soft robots to crawl and rotate flexibly (Fig. 6e, Supplementary Fig. 32 and Movie 16). Specifically, three H modules of the trefoil-shaped robot can drag it to crawl in three directions. Two adjacent modules can drag the robot towards any directions inside the angle formed by these two modules, by simply controlling the synthesis of the EVs. On the other hand, the tortoise-like robot has four H-module legs, and controlling two front modules to oscillate synchronously can make the robot crawl forwards, and vice versa. When the EVs of modules on the two sides are controlled to be opposite intermittently with 1-Hz and 1.0-A unipolar square-wave signal, the robot rotates clockwise around the center with an average speed of 5.8° s$^{-1}$. The rotation mechanism is similar to that shown in Supplementary Fig. 30a.

We also demonstrate an untethered padding soft robot as shown in Fig. 6f. This robot has a same configuration with the turning robot H||H (see Supplementary Fig. 30a), which can float on the water carrying a control unit integrated with power and signal controller module. The 4-Hz square-wave voltage signal induced synchronized swaying motion of the two inflated modules under the constant magnetic field can propel the robot on the water with an average speed of 34.98 mm s$^{-1}$ (1.06 BL s$^{-1}$) (Supplementary Fig. 33, Movie 17). Different from the above multimodal locomotion on a fixed plate magnet, we also demonstrate a soft underwater robot carrying two small pieces of tiny magnets onboard (Fig. 6g). The buoyancy provided by the two inflated H modules can balance part of the middle magnets' weight in water. A similar flapping motion as shown in Fig. 5a can be generated by the applied current to propel the robot in water with an average speed of 5.93 mm s$^{-1}$ (Supplementary Movie 18).

## Discussion

In this study, we introduce a new concept of self-vectoring control to achieve high flexibility and operational dimensionality for magnetic actuation and soft robots. Through reconfiguration of two elementary soft flux actuator modules (module V and module H), diverse SESRs with strip shapes, trefoil-like shapes and square shapes have been constructed, which demonstrate instant reprogrammable shape morphing or agile multi-modal locomotion in constant magnetic fields. The principle of self-vectoring control for SESRs are validated

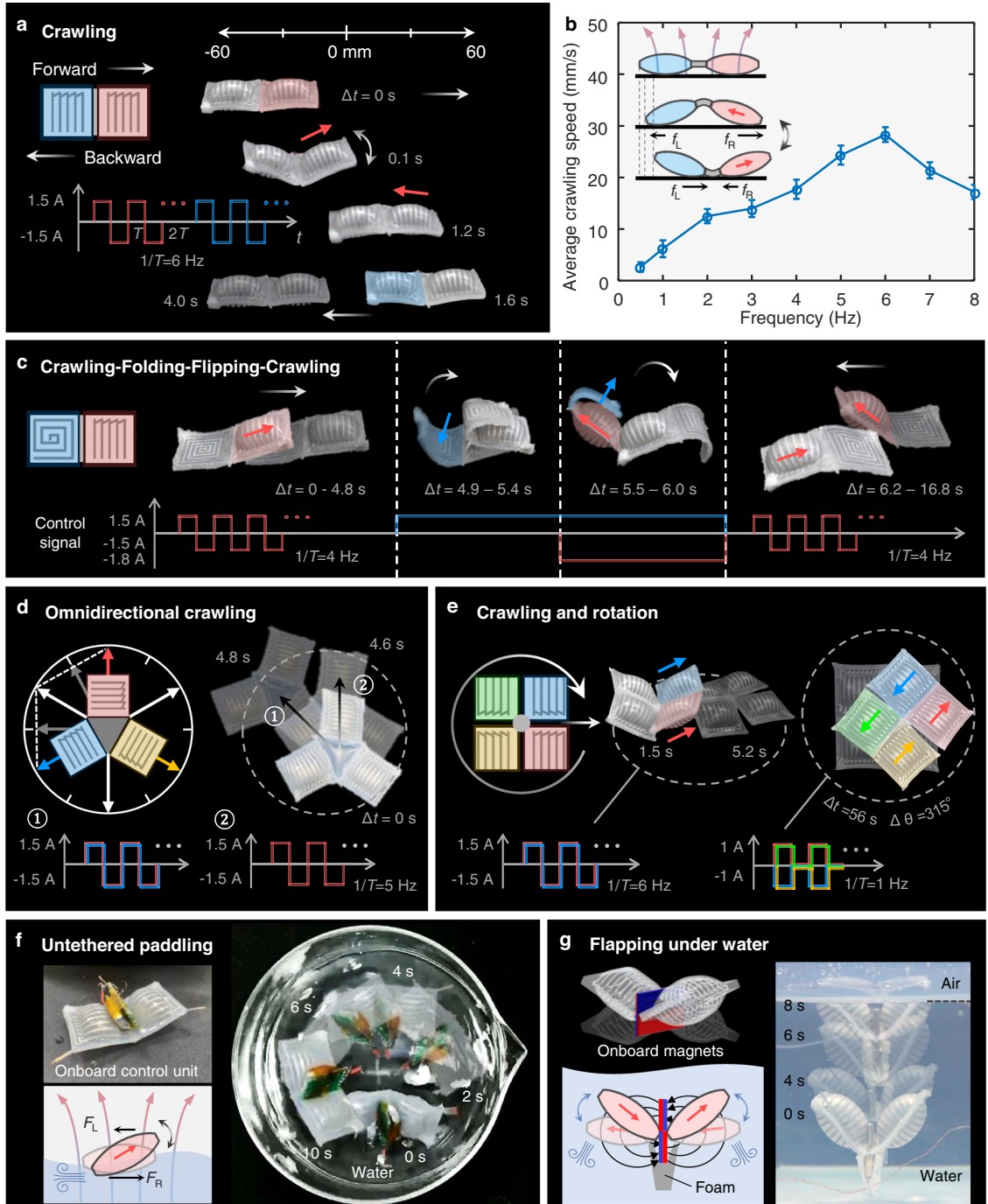

**Fig. 6 | Multimodal locomotion of SESRs by self-vectoring control.**
**a** Reciprocating crawling of an H-H robot with aligned EVs. Activating different module by a periodic bipolar square-wave signal, the robot crawls towards the corresponding direction like a caterpillar. **b** The crawling mechanism and the speed curve of the H-H robot. Error bars represent standard deviation. **c** Reciprocating crawling-folding-crawling motion of an H-V robot. The direction was changed by shape morphing and flipping movement. **d** Omnidirectional crawling of an H|| H||H robot. Any EV in the plane can be synthesized by the modules, thus to control any directional crawling. Two representative crawling directions are shown with the corresponding control signals. **e** Crawling and rotating of an H||H||H||H robot. The locomotion mechanisms and control signals are similar to the foregoing H-H robots. **f** Untethered paddling on water of a H||H robot. **g** Flapping under water of an H-H robot carrying with tiny magnets.

and demonstrated in different external constant magnetic fields. The actuation characterization of the unique module H actuator is investigated emphatically by theory, simulation and experiments, including the effects of air volume, inflation rate, and current signal.

The SESRs have shown the potential of operation in higher dimensionality with fewer actuators and control signals, such as the multi-modal and rhythmed dancing of the trefoil-shaped SESR and the agile rolling locomotion of the single or composite modules (Fig. 2c).

In addition, the diverse shape morphing and transformation (Fig. 5), along with the agile multimodal locomotion (Fig. 6), not only fully display the advantages of SESRs including fast response, easy reprogrammability, and selective actuation, but also verify the flexibility and scalability of the proposed concept of self-vectoring control.

Moreover, the instant reprogrammability simply controlled by current is much more flexible and convenient than the reprogrammable methods for soft magnetic machines[22–24]. The electrical

**Table 1 | Comparison of different soft actuators and robots on flexibility and operational dimensionality**

| Operation category | Actuation method (mode)* | Actuator No. | Signal No. | Reprogra- mmability | Selective actuation | Control complexity** | Respon- se time | Reference |
|---|---|---|---|---|---|---|---|---|
| Continuous rolling | Pneumatic | 8 | 8 | N | ✓ | Complex | s | [13] |
| | SMA | 7 | 7 | N | ✓ | Simple | s | [38] |
| | Magnetic field | 1 | External field | N | N | Complex | ms | [8] |
| | SESR | 1 | 1 | ✓ | ✓ | Simple | ms | This work |
| Omnidirect- ional motion | Pneumatic | 3 | 3 | N | ✓ | Complex | s | [34] |
| | LCE | 3 | 3 | N | ✓ | Simple | min | [39] |
| | Magnetic field | 1 | External field | N | N | Complex | ms | [19] |
| | SESR | 2 | 2 | ✓ | ✓ | Simple | ms | This work |
| Shape morphing | LCE (3 modes) | 48 | 4 | N | ✓ | Simple | min | [40] |
| | Magnetic field (1 mode) | 1 | External field | ✓ (complex and slow) | N | Complex | min | [22] |
| | Electromag- netic (1 mode) | 5 | 5 | ✓ | ✓ | Simple | ms | [28] |
| | SESR (3 modes) | 6 | 2 | ✓ | ✓ | Simple | ms | This work |

N no, ✓ yes.
*Mode means achievable shape morphing modes by a specific structure.
**The setup and accessory equipment for control system.

properties such as capacitance, inductance, and resistance of the LM coils in the actuators can also be used for the self-sensing of deformation and movement[32], which is still a challenge for the soft magnetic robots[33]. The comparison of different soft actuators and robots is listed in Table 1, which demonstrates the higher flexibility and operational dimensionality of SESRs for some operations. A more comprehensive comparison can be seen in Supplementary Table 4.

The SESRs proposed in this work may have some limitations, for requirements of tubes for inflating the module H, tethered electricity and a strong external magnetic field. Actually, the tubes are not mandatory, we can pre-store air in the chamber or inject a little low-boiling liquid to realize delayed inflation by phase transition. So, the module H can be rolled up and deployed for operation, which can have the same initial thin profile as module V (Supplementary Fig. 34, Movie 19). The untethered paddling robot with control unit onboard and the underwater swimming robot with tiny magnets onboard demonstrated in Fig. 6f, g can somehow solve the requirements of tethered electricity and a strong external magnetic field for SESRs, which show the potential application for underwater monitoring and exploration. Besides, the strong external magnetic field is available with MRI machines, which means the SESRs may be used for MRI-guided medical operations like positioning and mark[25,34]. The energy density, power density, and efficiency of soft electromagnetic actuators will also be greatly enhanced under such strong magnetic fields[28]. Of course, downsizing the modules, reducing the currents and improving the actuation performance can further expand the application potential of SESRs in the future[35,36].

In summary, new design and operation principle are reported here for providing an effective strategy to enrich the actuation modes and controllable dimensions for soft robots. It can be seen as increasing an active operational dimensionality for the soft magnetic robot system by instant reprogrammability and selective actuation. Specifically, we can customize external magnetic fields by permanent magnet array[37] or control spatial magnetic field gradient by Helmholtz coils to realize complex collaborative control with the SESRs[8]. The design concept and soft solution for the SESRs with instant reprogrammable shape morphing and multimodal locomotion capabilities will inspire diverse applications in medical robots, soft robotics, active metamaterials and human-machine interactions.

## Methods

### Materials

The SESRs were made of double-layered liquid metal micro-channels embedded in two patterned soft elastomeric shells. The implemented liquid metal consisted of gallium, indium, and tin with a mass ratio of Ga: In: Sn = (68.5: 21.5: 10) wt% (Santech, Hunan, China). The elastomeric shells were fabricated from Ecoflex 00-30 (Smooth-on, Pennsylvania, United States). In addition, the binder of the intermediate layers was Sil-Poxy (Smooth-on, Pennsylvania, United States).

### Fabrication and assembly of modules

Initially, all actuators were modeled using SolidWorks. Then, the additively manufacturing of the microchannel templates were based on an Objet30 Connex3 PolyJet 3D printer (Stratasys, Minnesota, United States). After dispensing the required amounts of Parts A and B into a mixing container (1 A:1B by weight or volume), to eliminate any entrapped air, we degassed the liquid rubber in the vacuum oven (DZF-6020AB, LICHEN, China). Finally, we cured the liquid Ecoflex in the 3D printed molds at room temperature (25 °C) for 2-4 h before demolding (or in the oven at 65 °C for 10 min, depending on the purpose).

After pasting the adhesive tape at the reserved position, we evenly brushed the binder to the inner side of the elastomer shell. Then, two membranes were closely bonded. For module H, a soft air tube made of silicone (Φ 0.3*0.8 mm, CS356, PureShi, Shanghai, China) was placed between the shells of the chamber and located near one corner of the module to connect the chamber with the air outside. The tube was glued using Sil-Poxy for obtaining good soft sealing. It takes 10-20 min for the binder to cure at room temperature fully. Next, the liquid metal (LM, EGaInSn) was injected into the double-layered microchannels using a syringe. When the needle was pulled out, a fine silver-plated wire with high temperature resistance (AF200, <200 °C, Jiaqi, Shanghai, China) was inserted into the small hole and sealed by Sil-Poxy. Then the two wires were wound together as the twisted pair. The tube of the module H could be cut away after inflating and the remaining vent can be sealed with instantaneous adhesive (HJ-420, Hui Ju, China) or a stopper (tiny nylon stick with 0.5 mm-dimeter).

Diverse SESRs with strip shapes, trefoil-like shapes and square shapes were all assembled and bonded directly by the Sil-Poxy form the two elementary modules V and H. Some linking modules were also molding with Ecoflex and then bonded with actuator modules with Sil-Poxy.

## Characterization of the actuators

To maintain better tensile properties, we choose the Eco-flex type with Shore hardness 00-30 and tensile strength 200 psi. The shear modulus of Ecoflex is obtained by a simple tension test with a constant strain rate of 0.66%·s$^{-1}$ using universal material testing machine (Zwick, Germany). The geometry of specimen was set to be 3.2*6 mm which referred to the European Standard EN ISO 527-2:1996 (type 5 A). Fitting with the incompressible neo-Hookean hyperelastic model obtains the shear modulus of the Ecoflex, 42.07 kPa (see Supplementary Fig. 7a).

The internal pressures corresponding to different inflating volumes of the module H were measured by a pressure sensor (XGZP6847A, 0-10 kPa, CFSensor, China). The pressure sensor was connected to the module sample and a 10-ml injector with a three-way connection, as shown in Supplementary Fig. 8c, the maximum pressure is about 3.64 kPa for the 7-ml air volume. The profile dimension of the inflated module H was simply measured by recording the inflating process with a high definition camera (FDR-AX700, SONY, Japan) and then analyzing the side-view pictures corresponding to different inflating volumes. The inflation of the module was also simulated by Abaqus, the measured pressures corresponding to different air volumes were used as the load parameters during the simulation. Supplementary Fig. 8d shows the comparison of simulation and experimental results on the profile dimension for module H. A good agreement between them is displayed. The height $h$ of the profile increases from the initial 2.8 mm ($V_a = 0$ ml) to final 19.9 mm ($V_a = 7$ ml), it is about 7-fold increase.

The resistances of the module samples were measured by a Source Measure Unit (B2910BL, Keysight, United States), as shown in Supplementary Fig. 8a. Resistances corresponding to different inflating volumes were plotted in Supplementary Fig. 8b.

## Characterization of the actuation and motion

Two blocky magnets were placed opposite to each other with a distance of 50 mm using a 3D-printed fixture. Then the composite modules V||H and H||H were hung in the center of the magnetic field with un-stretchable thin thread for excluding the effects of gravity and friction, respectively. Two current signals were applied on the modules V and H or H and H to control their motions during the qualitative validating experiments of self-vectoring control for SESRs. To quantitatively characterize the relationship between the rotation angle and the control signal (waveform, inflation rate, current amplitude and frequency), the whole setup can be divided into three parts: electronic system, robotics system, motion capture system (Supplementary Fig. 21). In the electronic system, DC power sources (UTP1310, maximum 320 W, UNI-T, Guangdong, China) output constant current ($I$, 0-2 A); A signal generator (AFG3022C, Tektronix, Oregon, United States) provides variable frequency ($f$, 0.5-30 Hz); A control circuit (OPA549, ± 8 A, Texas Instruments, Texas, United States) allows the current to reverse as the waveform. The motion capture system was the same high definition camera.

## Numerical simulation of the EV synthesis

We directly link CAD Software Solidworks and finite element software COMSOL multiphysics to import the model and set the appropriate air domain. Through the AC/DC module, the EV and its surrounding magnetic field are simulated, and the coupling effect of the two modules is verified. The model of module H used for simulation is a simplified helical coil with uniform cross section.

## Characterization and numerical simulation of the Lorentz torque

During the measurements, modules were also hung in the center of the magnetic field same as the qualitative validating experiments for self-vectoring control. The angle between the magnetic field and the modules were changed by adjusting the relative position of the fixture

for magnets. A homemade lamina probe was mounted on the load cell (LSB 200, 20 g, FUTEK, United States) to measure the blocking force generated by the activated actuator modules (Supplementary Fig. 19). The Lorentz torque can be equivalent by multiplying the measured blocking force by the moment arm.

The LM helical coil in the inflated module H with different inflating volume is simplified to a helical coil with uniform elliptic cross-section in the simulation (Supplementary Fig. 18). The short axis of the cross-section varies with the inflating volume, which can be evaluated by the measured profile dimension of the inflated module (Supplementary Fig. 8d). The long axis of the cross-section and the length of the coil are consistence with the samples. The magnetic field in the simulation was set to be consistence with the practical experimental setup, including the dimension, parameters, and spacing of the two blocky magnets. The details of simulation and experiments of the Lorentz torque can be found in Supplementary Notes.

## Characterization of the magnets

The two blocky magnets (N40, Jiuci, Beijing, China) and the circular plate magnet (N52, Jiuci, Beijing, China) are all made of NdFeB and coated with nickel-copper-nickel. The dimension of the blocky magnets is 50*50*38 mm, and the diameter and height of the plate magnet are 120 and 40 mm, respectively. We characterized the magnet with a Gauss meter (HT201, range: 0-2 T, sensitivity: 0.01 mT, Hengtong, Shanghai, China). The measured magnetic field strength is used to fit the remanent magnetization. We used the commercial finite-element software COMSOL Multiphysics to obtain the magnetic field distribution. The simulation assumes that the magnet material is homogenous and axisymmetric in geometry. The characterization results of the two kinds of magnets can be seen in Supplementary Figs. 17 and 20.

## Onboard control unit

The control unit simply consists of a miniature polymer lithium cell (851316RH20, YANLLPOWER, China) and a commercial time delay module enabled by MOSFET (HX-1M-03A-01, Huarui Electronics, China). The polymer lithium cell has a capability of 80 mAh and maximal discharge rate of 20 C (1.6 A), with a 3.7-V nominal voltage and 3.0-V end-off voltage. The dimension of the cell is 8.5*13*16 mm, with a 2.2-g weight. The positive and negative pole ears are respectively made of aluminum and copper foils which are nonmagnetic material. The time delay module can generate PWM signals with variable amplitude, tunable frequency and duty cycle. The output voltage is consistent with the input power (3-5 V). The frequency can be tuned between 119880$^{-1}$-50 Hz. The dimension of the module is 26.92*17.78*7.5 mm, with a 1.9-g weight. The polymer lithium cell and the paddling robot were connected to the delay module by the circuit interfaces of '-POW + ', and '+OUT·' respectively as shown in Supplementary Fig. 33f. The total weight of the control unit is 4.15 g.

## Fabrication of the untethered paddling SESR

Two H modules were assembled in parallel by bonding their side edges using Sil-Poxy. After inflating them with 5-ml air respectively, the air tubes were cut off from the robots and two tiny nylon sticks were used to plug the remaining vents. Then the control unit was mounted at the middle of the robot by Sil-Poxy, between the two inflated modules. The total weight of the untethered paddling SESR is 11.62 g.

## Fabrication of the underwater SESR

Two H modules were assembled in series by bonding their side edges using Sil-Poxy. Then two pieces of tiny magnets (N38, Jiuci, Beijing, China) with the dimension of 30*10*3 mm and weight of 7.5 g were mounted on the up and bottom side of the robot. The magnets were fixed at the middle interface of the two modules, the attraction between them provides a convenient way for assembling them to the robot. The polarities of the two magnets are opposite on one side, thus

a magnetic field can be generated to pass through the side module. The directions of the magnetic field through the two modules are also opposite for this deployment. So the EVs of the two modules should be in the same direction for generating synchronized flapping motion. A piece of foam was also attached to the bottom magnet for increasing the buoyancy for the robot. Then adjusting the inflating volume for the modules to make the robot suspend in the water. The total weight of the underwater swimming SESR is 22.5 g (excluding the wires and tubes).

## Data availability

All data needed to evaluate the conclusions in the paper are present in the paper and/or the Supplementary Information. Additional data related to this paper may be requested from the authors.

## Code availability

The code used in this paper is available upon any reasonable request.

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

## Acknowledgements

We thank K. Ma and X. Li for discussions about electromagnetic actuators and liquid metal. This work was supported by the National Natural Science Foundation of China (Grants no. 12002204 (W.L.), 12032015 (W.Z.), 12121002 (W.Z.) and 12002201 (L.S.)), the Fundamental Research Funds for the Central Universities, China (Grant no. 22120220515 (W.L.)), the Science and Technology Innovation Action Plan of Shanghai, China (Grant no. 21190760100 (W.Z.)), and the Shanghai Sailing Program of Shanghai Science and Technology Committee, China (Grant no. 19YF1425000 (L.S.)).

## Author contributions

W.L. conceived the project and designed the actuators. W.L. and H.C. fabricated the actuators. W.L. and H.C. developed the theoretical and numerical model for the actuators and conducted the analysis. W.L. designed the experiments and built the experimental setup. W.L. and H.C. conducted the experiments. W.L. and H.C. analyzed the results and wrote the manuscript. Z.Y., F.F., X.G., Z.W., Q.G., J.X., G.M., L.S. and W.Z. disscused and revised the manuscript. L.S. and W.Z. supervised the research.

## Competing interests

The authors declare no competing interests.
