## [Peer Review File · Nature Communications]

Self-vectoring electromagnetic soft robots with high operational dimensionalityREVIEWERS' COMMENTS:

Reviewer #1 (Remarks to the Author):

This paper introduced a novel concept of self-vectoring control to realize the adjustments of interior electromagnetic vectors of the robots through the combination of horizontal electromagnetic vector and vertical vector. Then they proposed soft vectoring electromagnetic soft robots (SESRs), which achieved different locomotion modes, including bending, twisting, flapping, crawling, and turning. The content of the article is interesting. I would like to recommend it for publication after minor improvements.

The following questions are for the authors to consider:

1. From line 112, "arbitrary spatial vector can be synthesized by three orthogonal EVs". However, the presented SESRs had only two coils, which seems to be unable to achieve arbitrary space vectors.

2. In Fig. 2, the inflated tubes did not appear on the SESRs. I am wondering how the air was sealed inside.

3. Due to the limitation of the external magnetic field generated by a permanent magnet, the movement time of SESRs was quite short. For long-time operation, the heating of the coils may become one problem for long-distance movements. Besides, the restraints of inflation tubes and current signal control lines also affect its further applications. I suggest more discussion around these problems.

4. The dimensions of the permanent magnet were inconsistent. In line 502, its radius is 120 mm while in supplementary Fig. 13, its diameter is 120 mm. Please check it.

Reviewer #2 (Remarks to the Author):

This work proposes electromagnetic soft robotic modules which can achieve arbitrary actuation direction. I like the concept of active and instant adjusting and synthesizing the interior electromagnetic vectors. However, I still have some concerns about the practicality of the robot since no exciting application demonstrations are provided. There are some issues should be addressed.

1. Why design module H as an inflatable structure instead of stationary? How to evaluate the potential influences of the bulky dimension of the module (after inflation) on the flexibility of the robot?

2. One of the advantages of magnetic robots is their untethered actuation and control, which is also one of the reasons for their potential applications in the biomedical field or micromanipulation. However, the electromagnetic soft robot in this work needs both a wired power supply and pneumatic control, any advantages of this design or any specific application scenarios?

3. Despite the authors demonstrating the deformation and motion of the electromagnetic soft robot under magnetic field, these results appear to be achievable with existing magnetic and pneumatic techniques. The author emphasizes the module's instant programming capabilities but does not give exciting application demonstrations. This makes me worry about the practicality of the robot.

4. Experiments and video presentations of robots are cool, but as a scientific paper, there seems to be a lack of quantitative data description and characterization. How were the magnitude and range of the applied magnetic field chosen in each demonstration experiment? If the magnetic field is generated by permanent magnets, are the effects of magnetic attraction or magnetic gradients considered? What is the relationship between torque and magnetic attraction and applied magnetic field and current? Actual generatable force and torque are missing and they should be calculated with a normalized sample with real values.

5. Fig. 1a is hard to understand. Considering the wide readers of Nature Communications, it is recommended to modify the figure to be more accessible and to highlight the characteristics of the

work at the same time.

Reviewer #3 (Remarks to the Author):

In this work, a group of self-vectoring electromagnetic soft robots was built with high operational dimensionality. The demonstrated robots are interesting and the artwork, including figures and videos, is attractive. The mechanism of the robots is not new, especially one of the two types of module, module V, which has been well studied in reference 28. Module H is a new design. The combination of modules V and H should be the highlighted point of this work, but from the demonstrated experiments, I cannot notice the advantage of this design compared to the other magnetic robots such as this work <https://www.science.org/doi/abs/10.1126/scirobotics.aav4494>. The major issue of this study is the superficial analysis which only characterizes the robots with a few experiments and does not give a valuable design guide. The possible application and scalability of module H may be a problem in the future. Also, some other issues exist in this work. Detailed comments are listed below.

Major issues

1. It seems that reference 28 is an important reference related to this study. I have read through this reference as well. The module V is almost identical to the actuator in this reference which is fine. However, I don't understand why the current study identically copied the experimental description (line 490-493 in this study) of the elastomer characterization from ref. 28 (section, Characterization of the elastomer): "The shear modulus of Ecoflex is obtained by a simple tension test, with a constant strain rate of 0.66%/s. The specimen geometry can be found in the European Standard EN ISO 527-2:1996 (type 5A). Fitting with the incompressible neo-Hookean hyperelastic model obtains the shear modulus of the Ecoflex, 49.12 kPa." As the ref. 28 is not cited in this context, does it mean the authors conduct the same experiments? From my experimental experience, I can conclude that the current study did not conduct the experiments because it is almost impossible to have the same measured shear modulus in different experiments. The authors need to give a convincing explanation.
2. The author uses the air volume and flow rate to characterize module H which to me is not proper. Especially, the air volume is the volume injected into module H if I am right. However, the pressure is the direct factor in the deformation of module H. I would suggest using the pressure instead of the volume to characterize the module such as those in Fig. 4.
3. After going through the manuscript, the main new stuff maybe is module H. It is good to use pressurized air to fabricate these liquid metal channels. However, the large air volume also limited its application. The air also limits the deformation of the module and makes it rigid-like. In this study, module H only rotates and behaves more like a rigid body. The high-pressure air also squeezes the liquid metal channel. What is the variation of the electric resistance of module H vs. pressure? And how is the temperature distribution? Will the heat generated by the joule effect heat up the air and further inflate module H?
4. What is the magnetic torque of the module? And how the torque changed with air pressure and current is more important which is not investigated.
5. The authors analyze the electromagnetic vector of the module V||H and H||H which is a qualitative study. Such an analysis is superficial and the result is straightforward. No idea how much of the magnetic torque to the modules and cannot optimize the modules with such kinds of simulations.
6. The authors show a lot of locomotion but did not give enough analysis of the locomotion. For example, the locomotion of module H||H in Fig. 4c. How does it locomote? What's the highest speed and how it is performing compared to the other technology?
7. I am wondering whether could the underwater SESR really swim in the video. I am afraid the SESR is lifted up because of the heated-up air in the chambers. By the way, again, it would be important to know the magnetic torque of module V for future design.
8. I am wondering about the application of the module V-related robots. They should be difficult to minimize. Also, the large and curved body makes it difficult to find proper application. By the way, the application shown in supplementary Fig. 8 is not impressive and cannot show the advantage of such a soft robot. A demonstrated function would be preferred.

Minor issues

1. The structure and fabrication of modules H and V are clear. But the structure of their combination is not clear. How is the interface of the two modules (such as V||H in Fig. 5e) and the structure module H||H in Supplementary Fig. 8. It seems that the interface is very thick.
2. Please indicate the materials clearly in supplementary Fig. 4d and Fig. 5d. What is the material of the transparent rectangular frame between the two microfluidic channels? It looks not soft. Is it made of clear plastic?
3. In line 483, the abbreviation, EGaIn is used as liquid metal which is different from that in the Material section, Ga: In Sn (line 468). The correct abbreviation should be EGaInSn. From the composition, it can also be called Galinstan.
4. In line 92, the expression "MRI-like static magnetic field" is not proper. The magnetic field of a plate magnet is different from that of an MRI machine both in amplitude and distribution.
5. In line 456, the expression "significantly enrich" to me is overstated if considering the works in the literature about magnetic robots including the SESRs related.

RESPONSE TO REVIEWERS' COMMENTS

Title

Self-vectoring electromagnetic soft robots with high operational dimensionality

Authors

Wenbo Li^{1,2,†*}, Huyue Chen^{3,†}, Zhiran Yi¹, Fuyi Fang¹, Xinyu Guo¹, Zhiyuan Wu¹, Qihua Gao¹, Lei Shao^{3*}, Jian Xu², Guang Meng¹, Wenming Zhang^{1*}

Affiliations:

¹State Key Laboratory of Mechanical System and Vibration, School of Mechanical Engineering, Shanghai Jiao Tong University; Shanghai, 200240, China.

²School of Aerospace Engineering and Applied Mechanics, Tongji University; Shanghai, 200092, China.

³University of Michigan–Shanghai Jiao Tong University Joint Institute, Shanghai Jiao Tong University; Shanghai, 200240, China.

†These authors contributed equally to this work

*Corresponding author. Email: wenboli@tongji.edu.cn, lei.shao@sjtu.edu.cn, wenmingz@sjtu.edu.cn.

1. We thank the reviewers for their constructive feedback on our paper. We have substantially modified the manuscript based on this feedback.
2. Below are point-to-point answers to reviewers' comments and questions. Red indicates the original reviewer comment, and our response is in black. In addition, we have modified the original text; new text and edits are marked up with yellow highlights. We believe that the modified version is significantly improved.

Reviewers' comments:

Reviewer #1 (Remarks to the Author):

This paper introduced a novel concept of self-vectoring control to realize the adjustments of interior electromagnetic vectors of the robots through the combination of horizontal electromagnetic vector and vertical vector. Then they proposed soft vectoring electromagnetic soft robots (SESRS), which achieved different locomotion modes, including bending, twisting, flapping, crawling, and turning. The content of the article is interesting. I would like to recommend it for publication after minor improvements.

We thank the reviewer for the positive comments on our paper's novelty and quality. Our specific response to each of the comments is detailed below.

The following questions are for the authors to consider:

1. From line 112, "arbitrary spatial vector can be synthesized by three orthogonal EVs". However, the presented SERS had only two coils, which seems to be unable to achieve arbitrary space vectors.

As the reviewer mentioned, we have illustrated the principle of electromagnetic vector (EV) synthesis by three modules (two elementary modules: module V with a vertical EV, module H with a horizontal EV) in the modified **Fig. 1a**.

Fig. 1 Concept of self-vectoring and the elementary soft actuator modules of SESRs. a Schematic of active self-vectoring of the SESRs. The sphere space with gradient from blue to red represents the passive and constant external magnetic field. Three modules (two elementary modules: module V with a vertical EV, module H with a horizontal EV) in the sphere space can synthesize any EVs in theory.

Page 6: “We conceive two elementary modules with interior reprogrammable EVs, module V contains a vertical electromagnetic flux B_V , and a horizontal flux B_H corresponding to module H. As we know, two orthogonal EVs can generate any vector in a plane following the parallelogram law of vector synthesis. Further, arbitrary spatial vector $B(r, \theta, \varphi)$ can be synthesized by three orthogonal EVs in theory, as shown in Fig. 1a.”

2. In Fig. 2, the inflated tubes did not appear on the SESRs. I am wondering how the air was sealed inside.

We thank the reviewer for this comment. Actually the tubes are not mandatory, we can pre-store a volume of air in the chamber or inject different volumes through a connected tube whenever necessary. The tubes can be cut away after inflating and the remaining vent can be sealed with instantaneous adhesive (HJ-420, Hui Ju, China) or a stopper (tiny nylon stick with

0.5 mm-diameter). In some characterization experiments, we reserved the tubes to investigate the effects of inflation air volume on the actuation. We have added the details of fabrication and sealing method for the module H in the section of “**Methods**”.

3. Due to the limitation of the external magnetic field generated by a permanent magnet, the movement time of SESRs was quite short. For long-time operation, the heating of the coils may become one problem for long-distance movements. Besides, the restraints of inflation tubes and current signal control lines also affect its further applications. I suggest more discussion around these problems.

We thank the reviewer for pointing out issues towards real-world applications. Indeed, for all electromagnetic coils related systems, whether they are used to generate magnetic fields or perform as actuators, their heating must be considered. To address these points in the paper, we measured the temperature variation of the modules excited by different currents. The corresponding results are shown in **Supplementary Fig. 9**. The results show that after a period of time, the temperature tends to be stable, higher current and larger inflation volume (increased resistance) correspond to higher temperature. It is worth noting that the highest temperature for invasive medical devices should not exceed 44 °C to avoid damaging the surrounding tissues. Therefore, for our SESRs, as long as the current is below 1 A or short time actuation (1.5 A within 100 s), even as implanted devices, they can meet the requirements.

In addition, for electromagnetic robots, untethered locomotion and adaptability to unstructured terrains have always been grand challenges. Here, we built an untethered SESR integrated with a tiny battery and a small onboard circuit. As shown in **Fig. 6f** and **Supplementary Fig. 33**, an H||H SESR can paddling and swimming on the water without any air tubes or electric wires. The control signal was generated by the onboard control unit, the air was pre-stored in the modules to provide the buoyancy for the robot. We believe that such a demonstration can pave the way for future real-world applications.

Moreover, an underwater SESR with tiny magnets onboard are also demonstrated in **Fig. 6g**, which shows the potential to get rid of the static strong magnetic field for SESRs. **Figs. 6f** and **g** can somehow solve the requirements of tethered electricity and a strong external magnetic field for SESRs, which show the potential application for underwater monitoring and exploration. The relevant contents can be found in the revised manuscript.

Supplementary Fig. 9. Temperature measurement of module H.

Fig. 6 f Untethered paddling on water of a H||H robot. **g** Flapping under water of an H-H robot carrying with tiny magnets.

Supplementary Fig. 34. Untethered SESR integrated with a tiny battery and a small onboard circuit.

4. The dimensions of the permanent magnet were inconsistent. In line 502, its radius is 120 mm while in supplementary Fig. 13, its diameter is 120 mm. Please check it.

We thank the reviewer for pointing out this inconsistent parameter. We're sorry for the carelessness. The real diameter and height of the plate magnet are 120 mm and 40 mm, respectively. We have revised it in the “**Methods**” section (**Page 28**).

Reviewer 2

This work proposes electromagnetic soft robotic modules which can achieve arbitrary actuation direction. I like the concept of active and instant adjusting and synthesizing the interior electromagnetic vectors. However, I still have some concerns about the practicality of the robot since no exciting application demonstrations are provided. There are some issues should be addressed.

We thank the reviewer for the insightful and valuable comments, which have enabled us to improve the manuscript. In the revised manuscript, we demonstrate several very challenging applications, hoping that they would be attractive enough.

1. Why design module H as an inflatable structure instead of stationary? How to evaluate the potential influences of the bulky dimension of the module (after inflation) on the flexibility of the robot?

We would like to thank the reviewer for providing us this opportunity to explain our unique design in detail. As we have demonstrated that the outer profile of module H can be reconfigured by inflation, so as the embedded LM coil. This unique design provides a new way to construct a solenoid with intrinsically stretchable materials to form two orthogonal horizontal electromagnetic vectors, which is different from the reported soft electromagnetic actuators with planar spiral coils. Based on these two kinds of design, a complete soft solution for self-vectoring actuation and robots is established, thus enabling previously unachievable shape morphing and locomotion capabilities in soft magnetic and electromagnetic robots.

The module H has the initial quadrature shape same with the module V, which make it very convenient to assemble and reconfigure in parallel or in series with other modules. As shown in **Fig. 2a**, a module H actuator first stacks up with a module V or H to constitute two composite modules V||H and H||H, respectively. The two modules are bonded together after assembly, then inflating the module H to result in the bulging configurations. If the module H is an initial bulging configuration, we can easily image that the assembly would be very hard for the composite modules.

Page 9: “The quadrature shape of the two actuator modules allows them to assemble and reconfigure expediently in parallel or in series to form new modules and configurations.” “The two modules are bonded together after assembly, then inflating the module H to result in the bulging configurations.”

We also quantitatively verified that increasing the inflation volume of module H can improve Lorentz torque through experiments and simulations (details in **Figs. 3e-g**, and **Supplementary Fig. 18**). In addition, in nature, many animals can also reshape themselves to switch between

different modes. For example, the swim bladder can regulate the buoyancy of fish, and puffer will expand in response to stimuli. We show details in **Supplementary Mov. 12** and **Supplementary Mov. 19**, inflation can also help our SESRs adjust its floating and posture to meet the needs of different scenarios.

As for the properties of the robot after inflation, on the one hand, we need to clarify that the air pressure inside is pretty low (almost $< 3\text{kPa}$), which will not affect the flexibility of the soft robots. To further explain the compliance of the inflatable actuator to the external stress, we have added a set of pictures in **Supplementary Fig. 7**, including stretching, twisting and pressing the module H with 5 mL air. Moreover, we can see the inflated module can realize agile rolling on different rugged terrain (**Fig. 2b** and **Supplementary Fig. 13**), which mainly attributes to its bulging configuration with less resistance for rolling.

Supplementary Mov. 12. Padding of SESRs assembled by two H modules with parallel electromagnetic vectors.

Supplementary Mov. 19. Actuation strategy of pre-stored low boiling point fluid.

Supplementary Fig. 17. Mechanical properties of inflated module H.

ii Agile rolling locomotion

Fig. 2b Demonstration of the agile rolling locomotion of a single H module and its composite module.

Supplementary Fig. 13. Demonstration of the agile rolling on a rugged terrain simulated by crumpled paper of a single module H.

2. One of the advantages of magnetic robots is their untethered actuation and control, which is also one of the reasons for their potential applications in the biomedical field or micromanipulation. However, the electromagnetic soft robot in this work needs both a wired power supply and pneumatic control, any advantages of this design or any specific application scenarios?

We thank the reviewer for pointing out this necessary comparison. First of all, we admit that the magnetic robots are indeed the best methods for untethered actuation and control in the laboratory. However, towards medical or industrial scenarios, the most significant difficult is to build a large-scale controllable 3D magnetic field (such as the magnetic field covering large organs or the whole human body). We believe that the MRI scanner is an excellent platform, which can provide a unidirectional super-strong magnetic field (3~7 T), which will over-magnetize the traditional magnetic robots. Previous studies (Ref. 25 and 26) have proved that the electromagnetic micro-catheter is highly MRI-compatible, with smaller artifacts, better heat optimization, and less puncture damage. Therefore, the purpose of our self-vectoring electromagnetic robots is not to replace the magnetic robot, but to cooperate and jointly create a better human-machine interaction experience.

Actually the tubes are not mandatory, we can pre-store a volume of air in the chamber or inject different volumes through a connected tube whenever necessary. The tubes can be cut away after inflating and the remaining vent can be sealed. In some characterization experiments, we reserved the tubes to investigate the effects of inflation air volume on the actuation. We also provide another inflation strategy without tubes: by pre-storing low-boiling point fluid. **Supplementary Fig. 34** provides a concept demonstration for the potential bioengineering application. **Fig. 2b** and **Supplementary Fig. 13** also demonstrate the agile rolling locomotion on different rugged terrains of module H, which may also provide benefit for this potential medical application.

Supplementary Fig. 34. The potential bioengineering application for module H by combining with a micro-catheter.

a Schematic of the combination of module H and a micro-catheter performing the gastric treatment. Illustrations include a folded module H that slides from the microcatheter, sinks, floats, and actuates. **b** Rolling up approach and photograph for further miniaturization. **c** Actuation strategy of pre-storing low boiling point fluid. The phase-changing temperature of the Novec 7000 (3M, Unite States) is 35 °C, which can quickly reshape the profile of the soft electromagnetic actuator inside the human body without tubes. **d** Proof-of-concept demonstration. In warm water, module H can show fast shape morphing, states transforming, and powerfully padding.

In addition, we built an untethered SESR integrated with a tiny battery and a small onboard circuit. As shown in **Fig. 6f** and **Supplementary Fig. 33**, an H||H SESR can paddling and swimming on the water without any air tubes or electric wires. The control signal was generated by the onboard control unit, the air was pre-stored in the modules to provide the buoyancy for the robot. An underwater SESR with tiny magnets onboard are also demonstrated in **Fig. 6g**,

which shows the potential to get rid of the static strong magnetic field for SESRs. **Figs. 6f** and **g** can somehow solve the requirements of tethered electricity and a strong external magnetic field for SESRs, which show the potential application for underwater monitoring and exploration. The relevant contents can be found in the revised manuscript.

We have compared different soft actuators and robots in **Table 1** and **Supplementary Table 4**. Through some indicators (such as Actuator No., Signal No., etc.), our self-vectoring actuation and robots demonstrate higher flexibility and operational dimensionality in some operations.

Fig. 6f Untethered paddling on water of a H||H robot. **g** Flapping under water of an H-H robot carrying with tiny magnets.

Supplementary Fig. 34. Untethered SESR integrated with a tiny battery and a small onboard circuit.

3. Despite the authors demonstrating the deformation and motion of the electromagnetic soft robot under magnetic field, these results appear to be achievable with existing magnetic and pneumatic techniques. The author emphasizes the module's instant programming capabilities but does not give exciting application demonstrations. This makes me worry about the practicality of the robot.

We thank the reviewer on the question of a comparison of our electromagnetic robots to previous magnetic robots. As for the traditional magnetic actuators, the problem of MRI-compatibility and selective actuation are still challenge. Moreover, the embedded magnetic particles cannot change their orientation, which named as “pre-programmable” (Ref. 8-10, 19-21). Therefore, their high motility completely depends on the changing magnetic field formed by multiple groups of large electromagnetic coils outside. Although there are also methods to change the direction of embedded magnetic particles, they always require a series of operations (heating and remonetizing) to complete the “re-programming process” (Ref. 22-24). The limitations are that they can only use materials with lower Curie temperature (for example, much weaker magnetism CrO₂), and the time-consuming process cannot achieve dynamic transformation. Let alone the response speed and frequency of the pneumatic actuators are far lower than the existing magnetic machines.

In this work, we propose a self-vectoring soft robot based on Lorentz force and vector synthesis, which means that even in a constant magnetic field like the MRI machines, our robots can also complete three-dimensional immediate shape morphing and multi-directional locomotion (as shown in the revised **Fig. 2b**). The SESRs have shown the potential of operation in higher dimensionality with fewer actuators and control signals, such as the multi-modal and rhythmmed dancing of the trefoil-shaped SESR and the agile rolling locomotion of the single or composite modules. The ‘leaves’ of the trefoil-shaped SESR can realize three modes of shape morphing including bending, twisting, bending and twisting by only two control signals. A single module H can act as a rolling soft robot which can be controlled by only one current signal to roll continuously on different rugged terrains (also see **Supplementary Fig. 13**). Then a composite module H||H can demonstrate the omnidirectional rolling controlled by only two current signals. We can see that synthesis and reprogramming of EVs controlled by only two current signals can lead to more than two rolling directions and shape morphing modes for the SESRs, which demonstrate a higher operation dimensionality compared with other soft actuation methods as listed in **Supplementary Table 4**.

Along with the concept demonstration of the potential bioengineering application, the untethered paddling SESR and the underwater SESR which have already been introduced before, all can give us the possibility and imagination for the practical applications. We believe that the instant reprogrammability of 3D shape morphing will also attract wide attention in the field of active metamaterials and human-machine interactions.

b Self-vectoring control under a static magnetic field

i High-dimensional shape morphing

ii Flexible rolling locomotion

Fig. 2. b i. High-dimensional shape morphing of a trefoil-shaped SESR assembled by three composite modules V||H in parallel. **ii.** Demonstration of the agile rolling locomotion of a single H module and its composite module.

4. Experiments and video presentations of robots are cool, but as a scientific paper, there seems to be a lack of quantitative data description and characterization. How were the magnitude and range of the applied magnetic field chosen in each demonstration experiment? If the magnetic field is generated by permanent magnets, are the effects of magnetic attraction or magnetic gradients considered? What is the relationship between torque and magnetic attraction and applied magnetic field and current? Actual generatable force and torque are missing and they should be calculated with a normalized sample with real values.

We thank the reviewer for the comments. We have added a lot of quantitative characterizations including experiment, simulation and analytical theory on the actuation performance of the modules under two different kinds of constant magnetic fields. The corresponding contents have also been demonstrated in the revised version.

First, we chose two kinds of constant magnetic fields generated by different magnets in the experiments. One is an approximately parallel magnetic field between two opposite blocky magnets as shown in **Fig. 3, Supplementary Figs. 16 and 19**. Two opposite magnets (N40,

Jiuci, Beijing, China) with different magnetic poles are fixed with a distance of 50 mm by customized fixtures to form a nearly parallel magnetic field. The corresponding characterization of this magnetic field is described in “**Methods**” section and shown in **Supplementary Fig. 17**. This magnetic field was mainly chosen for the qualitative verification of self-vectoring control (**Figs. 3 c and d**), and the quantitative measurements of Lorentz torque for the modules (**Figs. 3 e-f and Supplementary Fig. 18**). During the experiments, different modules were suspended in the center of the magnetic field by un-stretchable thin threads for excluding the effects of gravity and friction. Thus the modules can rotate freely under the action of Lorentz torque. The dimension of the magnets (50*50*38 mm) and their distance (50 mm) were chosen to be larger than the modules (33 mm side length), thus resulting an approximately parallel magnetic field passing through the modules and allowing their free motion in the space. Actually, larger dimension and stronger remanent magnetization of the magnets can give us a more ideal parallel magnetic field. But considering the fixation and the volume of the experimental setup, we chose the magnets in the experiments with the basic requirement. We measured the Lorentz torque of the modules varying with current and inflation volume. The corresponding simulations was also carried out, which show good agreement with experimental results. The deviations may mainly come from the simplified simulation model of the LM coil and the measuring error. The details are described in the “**Methods**” section and the **Supplementary Text**.

Page 11: “Figs. 3 e-g shows the comparison of experiment and simulation results on the Lorentz torque of the two elementary modules in the same magnetic field. When the direction of the coil section (following the right hand rule) is perpendicular to the constant magnetic field, we can obtain the maximum torque corresponding to different currents as shown in the figures. The torque increases approximately linearly with the current, and for module H, larger inflating volume results bigger torque. The simulations show good agreement with the experimental results. The deviations may mainly come from the simplified simulation model of the LM coil and the measuring error. We also evaluated the Lorentz torque when the modules were hung in the magnetic field at 45° angle (see Supplementary Fig. 18). The detailed methods of simulation and characterization can be seen in Supplementary Text and Supplementary Fig. 19.”

Page 28: “**Characterization and numerical simulation of the Lorentz torque.** During the measurements, modules were also hung in the center of the magnetic field same as the qualitative validating experiments for self-vectoring control. The angle between the magnetic field and the modules were changed by adjusting the relative position of the fixture for magnets. A homemade lamina probe was mounted on the load cell (LSB 200, 20g, FUTEK, United States) to measure the blocking force generated by the activated actuator modules (Supplementary Fig. 19). The Lorentz torque can be equivalent by multiplying the measured blocking force by the

moment arm.

The LM helical coil in the inflated module H with different inflating volume is simplified to a helical coil with uniform elliptic cross-section in the simulation (Supplementary Fig. 18). The short axis of the cross-section varies with the inflating volume, which can be evaluated by the measured profile dimension of the inflated module (Supplementary Fig. 8d). The long axis of the cross-section and the length of the coil are consistency with the samples. The magnetic field in the simulation was set to be consistency with the practical experimental setup, including the dimension, parameters, and spacing of the two blocky magnets. The details of simulation and experiments of the Lorentz torque can be found in Supplementary Text.”

Fig. 3 Validation and characterization of self-vectoring control for SESRs.

Supplementary Fig. 16. Experimental setup for validating the orientation control or self-vectoring control.

Supplementary Fig. 17. Characterization of the magnetic fields produced by two permanent magnets.

Supplementary Fig. 18. Experiment and simulation of the Lorentz torque of modules H and V.

Supplementary Fig. 19. Characterization of the Lorentz torque.

The other magnetic field was provided by a circular plate magnet with the dimension of $\Phi 120 * 40$ mm. Most of the shape morphing and multimodal locomotion demonstration as shown in **Figs. 2, 5 and 6** were conducted using this magnetic field. An axisymmetric magnetic field can be generated by the plate magnet, which provides the possibility for deriving the analytical model for the modules' motion. And the larger diameter can provide enough area for morphing and moving demonstration. For characterization of the rotation angle, critical condition and subsequent locomotion, we further clarify the experimental setup and characterize the permanent magnet in **Supplementary Fig. 20 and 21**. We also derived the model of the rotation motion for the inflated module H when located on the center of the magnet. The details of theoretical modeling can be found in **Supplementary Text**. The revised **Fig. 4 and**

Supplementary Figs. 25 and 26 show the quantitative data about the actuation characterization of the module H.

Supplementary Fig. 20. Experimental setup and method for charactering actuators.

Supplementary Fig. 21. Characterization of the plate permanent magnet and the actuation range of SESR.

Fig. 4 Actuation characterization of the module H under a constant magnetic field. e Force and motion analysis of the module H. f Comparison of theoretical and experimental results on the stable and critical rotation angle θ varying with the input current for different precharged air volume V_a .

Supplementary Fig. 25. Experimental results of steady rotation angle of the module H with different inflating volumes varying with the applied current.

Supplementary Fig. 26. Actuation characterization of the module H on a plate magnet.

Due to the fabricated materials for SESRs are nonmagnetic, and the power lines were also specially treated. Both ends of the double layered liquid metal coils in modules V and H are designed to converge at the same corner of the actuators and connect to the power source through soft twisted-pair wires. By this method, the effect of magnetic attraction on the actuators when subject to external magnetic field can be minimized. In addition, the magnetic gradients of the two kinds of magnetic fields were both taken into consideration in the simulation and theoretical model. The magnetic field in the simulation for Lorentz torque was set to be consistent with the practical experimental setup, including the dimension, parameters, and spacing of the two blocky magnets. And according to the characterization result of the plate magnet, as shown in **Supplementary Fig. 22**, we find a cylindrical region ($Z \leq 40$ mm and $R \leq 30$ mm) on the top of the magnet. The magnetic field is approximately parallel and decreases linearly along Z direction in this region, and the magnetic field magnitude in R direction is negligible. So the magnetic field in this region was analyzed with a linear fitting formula $B_z = -3.5711 * Z + 284.8$. We hope that these quantitative data description and characterization can better interpret our design and eliminate the questions of reviewers.

5. Fig. 1a is hard to understand. Considering the wide readers of Nature Communications, it is recommended to modify the figure to be more accessible and to highlight the characteristics of the work at the same time.

We thank the reviewer for the suggestion. We have modified **Fig. 1a** which directly demonstrates the electromagnetic vector (EV) synthesis by three modules (two elementary

modules: module V with a vertical EV, module H with a horizontal EV). Following the principle of vector synthesis, any EVs can be generated by the three modules in theory.

Fig. 1 Concept of self-vectoring and the elementary soft actuator modules of SESRs. a Schematic of active self-vectoring of the SESRs. The sphere space with gradient from blue to red represents the passive and constant external magnetic field. Three modules (two elementary modules: module V with a vertical EV, module H with a horizontal EV) in the sphere space can synthesize any EVs in theory.

Reviewer 3

In this work, a group of self-vectoring electromagnetic soft robots was built with high operational dimensionality. The demonstrated robots are interesting and the artwork, including figures and videos, is attractive. The mechanism of the robots is not new, especially one of the two types of module, module V, which has been well studied in reference 28. Module H is a new design. The combination of modules V and H should be the highlighted point of this work, but from the demonstrated experiments, I cannot notice the advantage of this design compared to the other magnetic robots such as this work <https://www.science.org/doi/abs/10.1126/scirobotics.aav4494>. The major issue of this study is the superficial analysis which only characterizes the robots with a few experiments and does not give a valuable design guide. The possible application and scalability of module H may be a problem in the future. Also, some other issues exist in this work. Detailed comments are listed below.

We thank the reviewer for carefully reading our manuscript and for the positive comments. We thank the reviewer for the insightful and valuable comments, which have enabled us to improve the manuscript.

To date, the previous soft electromagnetic actuators had only one vector form perpendicular to the plane (**ref. 28 and 29**), which severely restricted their maneuverability and imagination. Thus, the electromagnetic robots cannot achieve the 3D transforming and multimode locomotion compared to the magnetic machines. Here, we create an unprecedented approach to construct two orthogonal horizontal electromagnetic vectors with intrinsically stretchable materials. This technology fills the gap in the field of electromagnetic transducers, thus enabling arbitrary 3D orientation with the assembly of modular units (H & V). Therefore, our SESRs can not only complete instant shape transformation, but also evolve multimodal locomotion (even get rid of air tubes, wired power sources, and external magnetic field).

Moreover, the design and mechanism in **ref. 28** are not new either according to the reviewer's comment, for the planar spiral LM coil is a classic design which can be also found in **ref. 29 and 30**. But **ref. 28** gives us an improved design and effective analysis method for such soft electromagnetic actuators. We further improved the design for the module V in our work. Both ends of the double layered liquid metal coils were designed to converge at the same corner of the actuators and connect to the power source through soft twisted-pair wires. By this method, the effect of the tethered control on the actuators when subject to external magnetic field can be minimized (**Supplementary Figs. 1 and 2**). That's very important for the module to move in a constant magnet, which hasn't been involved in all previous work. We believe that this work will be an important progress in Lorentz force-based soft robots.

Compared to **ref. 21** in our manuscript (T. Xu, J. Zhang, *et al.*, *Sci. Robot.* 4, eaav4494 (2019)), we think that the two work are significantly different in the underlying logic. As for **ref. 21**, the authors emphasized a discrete 3D magnetization through locally UV exposure, which was regarded as a milestone in magnetic robots. However, once fabricated, embedded heterogeneous particles cannot be reprogrammed in situ (including strength, distribution, and orientation); Limited by the viscosity of precursor, the concentration of magnetic particles is not higher than 1:1; More importantly, the actuating principle depends on the time-varying external magnetic field, rather than passive and constant magnetic field. These problems are also the purpose of this paper.

Page 1: “Here, we report self-vectoring electromagnetic soft robots (SESRs) to offer new operational dimensionality via actively and instantly adjusting and synthesizing the interior electromagnetic vectors (EVs) in every flux actuator module of the robots. As a result, we can achieve high-dimensional motion and operation with fewer actuators and control signals than other actuation methods. We also demonstrate complex and rapid 3D shape morphing, bioinspired multimodal locomotion, as well as fast switches among different locomotion modes.”

Page 6: “Magnetic actuation is usually implemented through manipulating permanent magnets, electromagnetic coils, or electromagnets to generate spatially vectoring fields. The magnetic torques and forces exerted on magnetic robots can control their deformation and motion in the workspace. Here, we propose a distinction between external and interior magnetic actuation for vector control (Fig. 1a). We define the aforementioned magnetic actuation of changing the applied field direction as external vector control. The new concept of self-vectoring control is to actively and instantly adjust and synthesize the interior vectors in every sub-domain of the robots, thus to realize active and selective actuation in passive and constant fields.”

We also have added a lot of quantitative characterizations including experiment, simulation and analytical theory on the actuation performance of the modules under two different kinds of constant magnetic fields. The corresponding contents have also been demonstrated in the revised version.

The following is our point-to-point detailed reply:

Major issues

1. It seems that reference 28 is an important reference related to this study. I have read through this reference as well. The module V is almost identical to the actuator in this reference which is fine. However, I don't understand why the current study identically copied the experimental description (line 490-493 in this study) of the elastomer characterization from ref. 28 (section, Characterization of the elastomer): “The shear modulus of Ecoflex is obtained by a simple

tension test, with a constant strain rate of 0.66%/s. The specimen geometry can be found in the European Standard EN ISO 527-2:1996 (type 5A). Fitting with the incompressible neo-Hookean hyperelastic model obtains the shear modulus of the Ecoflex, 49.12 kPa.” As the ref. 28 is not cited in this context, does it mean the authors conduct the same experiments? From my experimental experience, I can conclude that the current study did not conduct the experiments because it is almost impossible to have the same measured shear modulus in different experiments. The authors need to give a convincing explanation.

We thank the reviewer for pointing out this and apologize for an oversight in the final check of the manuscript. Actually, we done the tension tests before. Because there are many soft materials involved in our research, we have characterized most of them and obtained the mechanical parameters, such as Ecoflex 00-30, 00-50, PDMS and some series of DragonSkin. Indeed, we referenced the writing of the section “Characterization of the elastomer” in ref. 28 for preparing the work, we forgot to amend the real parameter for our negligence. To remedy this error, we have added pictures and explanations in both “**Methods**” and “**Supplementary Materials**” in our revised manuscript:

Page 26: “The shear modulus of Ecoflex is obtained by a simple tension test with a constant strain rate of 0.66%/s using universal material testing machine (Zwick, Germany). The geometry of specimen was set to be 3.2*6 mm which referred to the European Standard EN ISO 527-2:1996 (type 5A). Fitting with the incompressible neo-Hookean hyperelastic model obtains the shear modulus of the Ecoflex, 42.07 kPa (see Supplementary Fig. 7a).”

Supplementary Fig. 7. Mechanical properties of Eco-flex 30.

a Measurement of Young’s moduli of the Eco-flex 30. **b** Stress-strain curve and local detailed data.

2. The author uses the air volume and flow rate to characterize module H which to me is not proper. Especially, the air volume is the volume injected into module H if I am right. However,

the pressure is the direct factor in the deformation of module H. I would suggest using the pressure instead of the volume to characterize the module such as those in Fig. 4.

We thank the reviewer for pointing out this important issue, and we also agree that it is necessary to provide the correspondence between air volume and pressure. Furthermore, we also established a physical model to predict the large deformation of soft materials during inflation through finite element simulation. The above materials have well verified the softness and compliance of module H after inflation, because the internal pressure is pretty low (<4 Kpa), which is different from the traditional pneumatic actuators (always > 100 kPa). We think the air volume is more intuitive for characterizing the profile variation of module H compared to the pressure.

Page 26: “The internal pressures corresponding to different inflating volumes of the module H were measured by a pressure sensor (XGZP6847A, 0~10 kPa, CFSensor, China). The pressure sensor was connected to the module sample and a 10-ml injector with a three-way connection, as shown in **Supplementary Fig. 8c**, the maximum pressure is about 3.64 kPa for the 7-ml air volume. The profile dimension of the inflated module H was simply measured by recording the inflating process with a high definition camera (FDR-AX700, SONY, Japan) and then analyzing the side-view pictures corresponding to different inflating volumes. The inflation of the module was also simulated by Abaqus, the measured pressures corresponding to different air volumes were used as the load parameters during the simulation. **Supplementary Fig. 8d** shows the comparison of simulation and experimental results on the profile dimension for module H. A good agreement between them is displayed. The height h of the profile increases from the initial 2.8 mm ($V_a=0$ ml) to final 19.9 mm ($V_a=7$ ml), it is about 7-fold increase.”

Supplementary Fig. 8. Mechanical, and geometric characteristics of module H with various air volumes.

3. After going through the manuscript, the main new stuff maybe is module H. It is good to use

pressurized air to fabricate these liquid metal channels. However, the large air volume also limited its application. The air also limits the deformation of the module and makes it rigid-like. In this study, module H only rotates and behaves more like a rigid body. The high-pressure air also squeezes the liquid metal channel. What is the variation of the electric resistance of module H vs. pressure? And how is the temperature distribution? Will the heat generated by the joule effect heat up the air and further inflate module H?

We thank the reviewer for the comments. We introduce the concept of "inflation" during fabrication to dynamically reshape the three-dimensional geometric profile of liquid metal channels. Nevertheless, at any time, we will not use high-pressure gas. As previously described, the highest pressure is always lower than 4 kPa. Therefore, the internal air only contributes to dynamically adjust the Lorentz torque of module H, and will not affect the overall flexibility.

Page 7: “The module H also has a very good compliance and robustness after inflation, which can be stretched, twisted, or pressed to a large extent (see **Supplementary Fig. 7**). The fabricated module samples have a good consistency, for their resistance and inflated outer profiles with different inflating volumes vary in small scale as shown in **Supplementary 8**.”

Supplementary Fig. 7. Mechanical properties of inflated module H.

Although we do not use high-pressure air, we agree that the resistance of liquid metals should be quantified as a function of the internal air pressure, as shown in **Supplementary Fig. 8a** and **b** (Refer to **Supplementary Fig. 8c** for conversion relationship between internal air pressure and volume).

Page 27: “The resistances of the module samples were measured by a Source Measure Unit (B2910BL, Keysight, United States), as shown in Supplementary Fig. 8a. Resistances corresponding to different inflating volumes were plotted in Supplementary Fig. 8b.”

Supplementary Fig. 8. Electrical characteristics of module H with various air volumes.

a Resistance measurement of the module H. **b** Resistance changes with various air volume. The pressure of filled air on the embedded microchannels will change the resistance, and the manual assembly will give the data a slight deviation between the three samples.

We also agree with the reviewer that the temperature distribution is not described clearly. According to this suggestion, we utilize infrared imaging to observe the overall thermal state, and flexible thermocouples to record dynamic thermal response data. After a period of time, the heating will tend to be stable. It is worth noting that the highest temperature for invasive medical devices should not exceed 44 °C to avoid damaging the surrounding tissues. Therefore, for our small-scale electromagnetic robots, as long as the current is below 1 A or short time actuation (1.5 A within 100 s), even as implanted devices, they can meet the requirements (**Ref. 25, 26**). In addition, in the future, we may be able to perform heat treatment to local lesions in fluids (Ref. J. Zhang, R. H. Soon, *et.al*, *Adv. Sci.* **2022**, 2203730).

Finally, the Joule heat effect will indeed heat the internal air, but the effect is very minute. The subsequent stable suspension in water for a long time can also illustrate this problem.

Page 7: “The temperature variations of the H samples with different inflating volumes under different current signals are also characterized, the results show that the temperature of the samples eventually tends to be stable (see Supplementary Fig. 9). The heating is slightly more pronounced for larger inflating volume, mainly caused by the increase of coil resistance with the inflating volume for module H.”

Supplementary Fig. 9. Temperature measurement of module H.

a, c, e Infrared images with different air volume and constant current. The reference room temperature is 23.5 °C, we selected the center part of the soft actuator with a square box to display the local maximum and minimum temperature. **b, d, f** Static temperature-time curve for the soft actuator subjected to different static currents, $I = 0.5, 1.0, 1.5 \text{ A}$. All plotting data are from the average temperature by infrared imaging, and marked with the local maximum temperature. **g** Experimental setup for dynamic temperature test (sampling frequency = 1 Hz). The flexible thermocouple is attached to the central area. **h** Dynamic temperature-time curve and local detailed data (dynamic frequency = 1 Hz).

4. What is the magnetic torque of the module? And how the torque changed with air pressure and current is more important which is not investigated.

We thank the reviewer for raising this excellent view and fully agree to improve torque-relevant sections. We have added a lot of quantitative characterizations including experiment, simulation and analytical theory on the Lorentz torque of the modules under different kinds of constant magnetic fields. The corresponding contents have also been demonstrated in the revised version.

Actual Lorentz torque are calculated with the normalized sample in both experiment and simulation. Both inflation and increased current strategies can significantly increase Lorentz torque, which will be helpful for the shape-morphing and locomotion.

Page 11: “Figs. 3 e-g shows the comparison of experiment and simulation results on the Lorentz torque of the two elementary modules in the same magnetic field. When the direction of the coil section (following the right-hand rule) is perpendicular to the static magnetic field, we can obtain the maximum torque corresponding to different currents as shown in the figures. The torque increases approximately linearly with the current, and for module H, larger inflating volume results bigger torque. The simulations show good agreement with the experimental results. The deviations may mainly come from the simplified simulation model of the LM coil and the measuring error. We also evaluated the Lorentz torque when the modules were hung in the magnetic field at 45° angle (see Supplementary Fig. 18). The detailed methods of simulation and characterization can be seen in Supplementary Text and Supplementary Fig. 19.”

Fig. 3 e f Comparison of experiment and simulation on the maximum Lorentz torque of the module H. The module is hung in the static magnetic field with different posture for e and f. **g** Comparison of experiment and simulation on the maximum Lorentz torque of the module V varying with applied current.

Supplementary Fig. 18. Magnetic torque validation of module H and V.

a to c Comparison of experimental and simulated magnetic torque, involving module H and module V in triaxial electromagnetic vectors at 45 degrees.

Supplementary Fig. 19. Characterization of the magnetic torque.

a Experimental setup for the magnetic torque tests. **b, c, d** Zoom-in details about module V, module H 90°, and module H 45°. Two opposite magnets with different magnetic poles are fixed on the 3D-printed shell and support system. All soft actuators are suspended to minimize the effects of gravity, friction, etc.

Page 28: “Characterization and numerical simulation of the Lorentz torque. During the measurements, modules were also hung in the center of the magnetic field same as the qualitative validating experiments for self-vectoring control. The angle between the magnetic field and the modules were changed by adjusting the relative position of the fixture for magnets. A homemade lamina probe was mounted on the load cell (LSB 200, 20g, FUTEK, United

States) to measure the blocking force generated by the activated actuator modules. The Lorentz torque can be equivalent by multiplying the measured blocking force by the moment arm.

The LM helical coil in the inflated module H with different inflating volume is simplified to a helical coil with uniform elliptic cross-section in the simulation (**Supplementary Fig. 18**). The short axis of the cross-section varies with the inflating volume, which can be evaluated by the measured profile dimension of the inflated module (**Supplementary Fig. 8d**). The long axis of the cross-section and the length of the coil are consistent with the samples. The magnetic field in the simulation was set to be consistent with the practical experimental setup, including the dimension, parameters, and spacing of the two block magnets. The details of simulation and experiments of the Lorentz torque can be found in **Supplementary Text.**

We hope that these quantitative data description and characterization can better interpret our design and eliminate the questions of reviewers.

5. The authors analyze the electromagnetic vector of the module $V||H$ and $H||H$ which is a qualitative study. Such an analysis is superficial and the result is straightforward. No idea how much of the magnetic torque to the modules and cannot optimize the modules with such kinds of simulations.

We thank the reviewer for pointing out this important issue, and we also agree that it is necessary to explain it in greater detail. The magnetic torque based on Lorentz force is a key performance index. We have quantitatively verified how the inflation volume (air pressure), current and angle are related to the torque through experiments. As shown in **Supplementary Fig. 17**, we carry out finite element simulation of the magnetic field, and calibrate it through experimental data. Subsequently, a series of Lorentz torque can be obtained through multi physical field simulation. In **Supplementary Fig. 18 d-f**, we also visualize the corresponding Lorentz torque. Different arrows represent the direction and amplitude of the electromagnetic vector generating Lorentz torque. To address this issue, we added “Simulation of the Lorentz torque” and “Characterization of the Lorentz torque” in **Supplementary Text**.

Page 9: “As we know, current-carrying wires are subjected to Lorentz force under a magnetic field. While for a closed loop of wire, a net Lorentz torque can be induced. So the actuators deform and move under the action of the driving force $F=I \cdot dl \times B$ or the torque vector $T=nI \cdot (A \times B)$, where I is the applied current, dl is an infinitesimal length vector of the coil, n is the number of coil loops and A is the area plane vector of the loops (25).”

Supplementary Fig. 17. Characterization of the magnetic fields produced by two permanent magnets.

Supplementary Fig. 18. Magnetic torque validation of module H and V.

Further, we have expanded and modified the quantitative characterization of **Fig. 4**, **Supplementary Fig. 22-26**, including dynamic model, rotation angle, critical current, etc., to predict the active vector control of a single H module under the constant magnetic field, and establish closed loop to optimize our design. The detailed “theoretical analysis of the module H on a plate magnet” is included in the **Supplementary Text**. The magnetic gradient and real profiles of the modules and LM coils with different inflation volumes were considered in the established model. We succeed to predict the tendency of the rotation angle varying with current and inflation volume, also the critical currents for flipping. This model also provides us an effective way to calculate the Lorentz force and torque for a complex structure.

Page 13: “**Fig. 4a-d** demonstrate four different modes of motions for the module (**Supplementary Movies 4 and 5**). They are corresponding to different inflating rates of air and current signals. Specifically, **Fig. 4a** is the inflating and rotation process when the module is inflated by different air flow under a certain current. **Fig. 4b** is the side to side swaying motion when the module is subject to different AC current signals with slow loading rate. **Fig. 4c** shows the side to side flipping motion when the module is excited by periodic pulse or step current signals. The EV flips together with the module, thus the periodic current with unipolarity can induce alternating clockwise and counterclockwise Lorentz torques which drive the module to flip side to side on the magnet. However, if changing the current polarity in the signal period, the direction of the EV reverses from right to left as the initial state, then a same clockwise torque makes the module flip right again to realize a continuous flipping motion as shown in **Fig. 4d**.”

Page 14: “**Fig. 4e** shows the force and motion analysis of the module H. The rotation motion is determined by a combination of Lorentz forces (F_{Li} and F_{Ri}), Gravity (Mg), reaction force (F_N), and frictional force (f). The detailed analysis and derived process are given in Supplementary Text. We obtain the theoretical relation between the rotation angles and applied current which is also verified by experiments as shown in **Fig. 4f**. The rotation angle of the inflated module first increases with the current, and larger inflating volume corresponding to larger angle under the same current. Then due to the practical profile of the inflated module is different from the hypothetical ellipse, the convex parts on both sides lead to the change of tendency on the rotation. When the endpoint of the convex part begins to contact the surface, the rotation angle will stay at this critical angle θ_c with the increasing of current, until big enough current rotates the module again. However, during the practical rotation process, the soft convex part deforms and bends under the gravity, which is different from the theoretical analysis. More experimental results about the steady rotation of the module H can be seen in **Supplementary Fig. 25**.”

Fig. 4 Actuation characterization of the module H under a static magnetic field.

Supplementary Fig. 22. The force analysis for the module H rotating under a static magnetic field.

Supplementary Fig. 23. The force analysis for the module H rotating under a static magnetic field.

Supplementary Fig. 24. The transition state for different rotation tendencies of the module H.

6. The authors show a lot of locomotion but did not give enough analysis of the locomotion. For example, the locomotion of module HIIHIIH in Fig. 4c. How does it locomote? What's the highest speed and how it is performing compared to the other technology?

We thank the reviewer for this comment. We have illustrated the crawling mechanism and crawling performance varying with frequency in **Fig. 6**. We also added discussions on the crawling mechanism and their speeds at different frequencies. This comprehensive analysis can help us obtain the best actuating parameters, and also let readers understand our design. In addition, the mechanisms of paddling on water and flapping under water are also shown in the illustrations. We have to admit that the demonstrated locomotion speed does not have obvious advantages so far, but it's acceptable for some potential applications. We think downsizing the modules, reducing the currents and improving the actuation performance can further expand the application potential of SESRs in the future.

Page 19: “**Fig. 6a** shows two H modules are connected in series to constitute a crawler. Applying a bipolar square-wave signal to one module can make it oscillate side to side similar to the flapping motion shown in Fig. 5a. The other module acts as a passive and compliant tail which can convert the oscillation of the robot to the caterpillar-like crawling by the differential friction. Therefore, controlling the oscillation for different module H can make the robot crawl towards two directions (**Supplementary Movie 11**). The crawling mechanism and the speed varying with driven frequency are depicted in **Fig. 6b**. There is a maximum crawling speed ~ 28.8 mm/s corresponding to the 6-Hz and 1.5-A control signal.”

Fig. 6 Multimodal locomotion of SESRs by self-vectoring control. a Reciprocating crawling of an H-H robot with aligned EVs. Activating different module by a periodic bipolar square-wave signal, the robot crawls towards the corresponding direction like a caterpillar. **b** The crawling mechanism and the speed curve of the H-H robot.

Page 21: “A positive or negative square-wave signals, the robot can turn clockwise or anticlockwise with an average speed of $4.7^\circ/s$. (**Supplementary Fig. 30a, Movie 12**). Further, we demonstrate the crawling and turning locomotion with the flexible transition by a soft robot

composed by three H modules (Supplementary Fig. 30b, Supplementary Movie 13). Actuating the upper H module or the bottom two modules H can realize crawling or bi-directional turning in one robot, respectively.”

Supplementary Fig. 30. Crawling and turning locomotion of an H-H robot.

It can be seen as the combination of the two H-H robots with crawling and turning locomotion capacities respectively. When crawling, the upper H module oscillates periodically and acts as a traction head to drive the whole robot to crawl forward. While for turning, the bottom two serial H modules are activated with opposite currents, the modules then sway to drive the robot turning clockwise or counter-clockwise under corresponding control signals.

Fig. 6f Untethered paddling on water of a H-H robot. **g** Flapping under water of an H-H robot carrying with tiny magnets.

7. I am wondering whether could the underwater SESR really swim in the video. I am afraid the SESR is lifted up because of the heated-up air in the chambers. By the way, again, it would be important to know the magnetic torque of module V for future design.

We thank the reviewer for raising this interesting view, to be honest, we have discussed the physical mechanism of swimming robots before. As shown in **Supplementary Movie 18**, the underwater movement of SESRs is unique, and it is not as simple as changing the density and lifting up. Unlike previous underwater magnetic robots (such as Z. Ren, W. Hu *et al.*, *Nat. Commun.* **10**, 2703 (2019) and Ref. 8), they extremely rely on the variable magnetic field provided by the external electromagnetic coils; Here, an H-H robot carries tiny magnets, and breaks free from the restraint of external magnetic field, completing periodic flapping.

Inspired by fishes in nature, the fast swimming upward is actually the result of swinging fins, especially for species without bladder, such as sharks and mantas; The function of the bladder is to maintain the suspension in the water (there is no muscle around to actively regulate the size of bladder). Similarly, the levitation of SESRs in water needs to control the internal air volume, but Lorentz torque acts as the mastermind behind rapid swimming (8s). The experiment also fully proves our view that a long time of power on will indeed heat the internal air, but ultimately only slightly change the position of the suspension (90s).

We consider that such discussion and interpretation can be persuasive enough.

Fig. 6 g Flapping under water of an H-H robot carrying with tiny magnets.

Page 22: “Different from the above multimodal locomotion on a fixed plate magnet, we also demonstrate a soft underwater robot carrying two small pieces of tiny magnets onboard (**Fig. 6g**). The buoyancy provided by the two inflated H modules can balance part of the middle magnets’ weight in water. A similar flapping motion as shown in Fig. 5a can be generated by

the applied current to propel the robot in water with an average speed of 5.93 mm/s. (Supplementary Movie 18).”

Supplementary Mov. 18. Suspending of a SESR assembled by two H modules (without tiny magnets).

8. I am wondering about the application of the module V-related robots. They should be difficult to minimize. Also, the large and curved body makes it difficult to find proper application. By the way, the application shown in supplementary Fig. 8 is not impressive and cannot show the advantage of such a soft robot. A demonstrated function would be preferred.

We thank the reviewer for this comment. We guess the reviewer is concerning the module H-related robots. We fully agree to demonstrate a more complex and functional application scenario to demonstrate the superior property of our robots.

First, based on the existing process and actuators in this work, we provide a strategy of “miniaturization by rolling” and an inflation approach without tubes. In **Supplementary Fig. 33**, a demonstrated function is available towards bioengineering and medical application. Recently, liquid metals with excellent biocompatibility have been developed and promoted in succession (such as Y. Lu, Q. Hu *et al.*, *Nat. Commun.* **6**, 10066 (2015), K. Nan, S. Babae *et al.*, *Nat. Biomed. Eng.* **6**, 1092-1104 (2022)), which is expected to be alternative medical materials for highly-toxic NdFeB magnetic particles.

Supplementary Fig. 33. The potential bioengineering application for module H by combining with a micro-catheter.

a Schematic of the combination of module H and a micro-catheter performing the gastric treatment. Illustrations include a folded module H that slides from the microcatheter, sinks, floats, and actuates. **b** Rolling approach and photograph for further miniaturization. **c** Actuation strategy of pre-filling LBP (low boiling point) fluid. The phase-changing temperature of the Novec 7000 is 35 °C, which can quickly reshape the profile of the soft electromagnetic actuator inside the human body without tubes. **d** Proof-of-concept demonstration. In warm water, Module H can show fast shape morphing, states transforming, and powerfully padding.

Then, instant self-vectoring enables the robots to have higher operational dimensions. We demonstrate the continuous rolling on rugged terrains with single module H and the omnidirectional rolling of an H||H composite module.

Page 10: “The right part of **Fig. 2b** shows the agile rolling locomotion of a single H module and its composite module H||H. First a single module H acts as a rolling soft robot which can be controlled by only one current signal to roll continuously on different rugged terrains (also see **Supplementary Fig. 13, Movie 2**). Then a composite module H||H demonstrates the omnidirectional rolling controlled by only two current signals. Here we give eight representative synthesized rolling directions corresponding to eight pulse synthesized EVs with

opposite directions. In addition, programming eight groups of pulse current signals in a sequence can control the module to rotate around the center for a rotary lighting operation (Supplementary Figs. 14, 15 and Movie 3). We can see that synthesis and reprogramming of EVs controlled by only two current signals can lead to more than two rolling directions and shape morphing modes for the SESRs.”

Fig. 2 Different configurations of SESR prototypes and self-vectoring control under a static magnetic field. b (i). High-dimensional shape morphing of a trefoil-shaped SESR assembled by three composite modules V||H in parallel. Every ‘leaf’ can demonstrate three modes of deformations including bend, twist, bend and twist under control of two signals. Scale bar, 20 mm. **(ii).** Demonstration of the agile rolling locomotion of a single H module and its composite module. A single module H can roll continuously on rugged terrains controlled by only one current signal, scale bar, 20 mm. The composite module H||H can realize omnidirectional rolling on the magnet surface controlled only two current signals, scale bar, 30 mm.

Supplementary Fig. 13. Demonstration of the agile rolling on a rugged terrain simulated by crumpled paper of a single module H.

Finally, we proposed the untethered SESR integrated with a tiny battery and a small onboard circuit (without air tubes and the wired power), and a SESR assembled with onboard magnets (without external magnetic field) in **Fig. 6, Supplementary Mov. 17** and **Mov. 18**. The corresponding fabrication and method have been added in “Methods” section in the revised manuscript. We believe that such demonstrations can pave the way for future real-world applications.

Fig. 6 Multimodal locomotion of SESRs by self-vectoring control. f Untethered paddling on water of a H||H robot. **g** Flapping under water of an H-H robot carrying with tiny magnets.

Page 22: “We also demonstrate an untethered paddling soft robot as shown in **Fig. 6f**. This robot has a same configuration with the turning robot H||H (see **Supplementary Fig. 30a**), which can float on the water carrying a control unit integrated with power and signal controller module. The 4-Hz square-wave voltage signal induced synchronized swaying motion of the two inflated modules under the static magnetic field can propel the robot on the water with an average speed of 34.98 mm/s (1.06 BL/s) (**Supplementary Fig. 33, Movie 17**). Different from the above multimodal locomotion on a fixed plate magnet, we also demonstrate a soft underwater robot carrying two small pieces of tiny magnets onboard (**Fig. 6g**). The buoyancy provided by the two inflated H modules can balance part of the middle magnets’ weight in

water. A similar flapping motion as shown in Fig. 5a can be generated by the applied current to propel the robot in water with an average speed of 5.93 mm/s. (**Supplementary Movie 18**).”

Minor issues

1. The structure and fabrication of modules H and V are clear. But the structure of their combination is not clear. How is the interface of the two modules (such as V||H in Fig. 5e) and the structure module H||H in Supplementary Fig. 8. It seems that the interface is very thick.

We thank the reviewer for pointing out this necessary fabrication. To address these points in our manuscript, we added text to the “Method” section. Only a very thin layer of Sil-Poxy or semi-cured precursor is required to complete the bonding.

Page 26: “Diverse SESRs with strip shapes, trefoil-like shapes and square shapes were all assembled and bonded directly by the Sil-Poxy from the two elementary modules V and H. Some linking modules were also molding with Ecoflex and then bonded with actuator modules with Sil-Poxy.”

2. Please indicate the materials clearly in supplementary Fig. 4d and Fig. 5d. What is the material of the transparent rectangular frame between the two microfluidic channels? It looks not soft. Is it made of clear plastic?

We are very grateful to the reviewer for carefully reading our materials, and we should clarify that all modules do not contain any hard materials, completely composed of highly-stretchable Eco-flex 30 and liquid metal. To answer this question, we have added **Supplementary Fig. 5** for describing the fabrication and materials more clearly. There are two soft membranes in the middle, and the middle square area is not coated with gel. These edges are remained after demolding to assist our assembly. There is no clear plastic.

Supplementary Fig. 5. Components of elastomer shells and the inflating deformation of module H.

There are two groups of soft shells for constructing the module H. The outside elastomer shells contain the microchannels for flowing the LM, and the inside elastomer shells are the sealing layers for the outside shells and also the walls of the chamber. The LM lines in the microchannels of the outside shells are connected through the abreast holes of the inside shells to form the interconnected LM coil. The module H is soft and its profile can be reconfigured by inflation, as well as the embedded LM helical coil.

3. In line 483, the abbreviation, EGaIn is used as liquid metal which is different from that in the Material section, Ga: In Sn (line 468). The correct abbreviation should be EGaInSn. From the composition, it can also be called Galinstan.

We thank the reviewer for this comment. We have changed the text accordingly.

Page 25: “the liquid metal (LM, EGaInSn)”.

4. In line 92, the expression “MRI-like static magnetic field” is not proper. The magnetic field of a plate magnet is different from that of an MRI machine both in amplitude and distribution.

We thank the reviewer for this comment. The magnetic field of MRI mentioned here means that its main magnetic field is unidirectional and static, which is not applicable to traditional magnetic robots. As for our SESRs, we think MRI scanner is an excellent platform, which can provide a unidirectional super-strong magnetic field (3~7 T). Previous studies (**Ref. 25 and 26**)

have proved that the electromagnetic micro-catheter is highly MRI-compatible. At the same time, the stronger magnetic field can significantly reduce the actuating current, which means that we can combine SESRs with much smaller batteries and even wireless charging or ultrasonic transducers. We also have removed this sentence for a more rigorous expression in the revised manuscript.

5. In line 456, the expression “significantly enrich” to me is overstated if considering the works in the literature about magnetic robots including the SESRs related.

We thank the reviewer for this comment. We have compared different soft actuators and robots on the flexibility and operation dimensionality in **Table 1**, and a more comprehensive comparison can be seen in **Supplementary Table 4**. Through some indicators (such as Actuator No., Signal No., Reprogrammability, Selective actuation, Control complexity and Response time), our self-vectoring actuation and robots demonstrate the higher flexibility and operational dimensionality in some operations. We also have modified the expression for a more rigorous point of view in the revised manuscript.

Page 24: “In summary, new design and operation principle are reported here for providing an effective strategy to enrich the actuation modes and controllable dimensions for soft robots.”

REVIEWERS' COMMENTS

Reviewer #1 (Remarks to the Author):

OK

Reviewer #3 (Remarks to the Author):

A lot of improvement has been made in this version. Even though the demonstrated application does not seem to be promising in the future practical application from my personal view. It is acceptable to be considered for publication.

RESPONSE TO REVIEWERS' COMMENTS

Reviewer #1 (Remarks to the Author):

OK

Response: We thank the reviewer for the positive comments on our paper.

Reviewer #3 (Remarks to the Author):

A lot of improvement has been made in this version. Even though the demonstrated application does not seem to be promising in the future practical application from my personal view. It is acceptable to be considered for publication.

Response: We thank the reviewer for the positive comments on our revised manuscript and the recommendation. The insightful and valuable comments from the reviewer have enabled us to improve the manuscript a lot. We'll further improve our work and explore more exciting applications in the future.